# Cross-Diffusion Waves resulting from Multiscale, Multi-Physics Instabilities: Theory

Klaus Regenauer-Lieb[1], Manman Hu[2], Christoph Schrank[3], Xiao Chen[1], Santiago Peña Clavijo[1], Ulrich Kelka[4], Ali Karrech[5], Oliver Gaede[3], Tomasz Blach[1], Hamid Roshan[1], and Antoine B. Jacquey[6]

[1]School of Minerals and Energy Resources Engineering, UNSW, Sydney, NSW 2052 Australia
[2]Department of Civil Engineering, The University of Hong Kong, Hong Kong
[3]Science and Engineering Faculty, Queensland University of Technology, Brisbane, QLD, 4001, Australia
[4]CSIRO, ARRC Technology Park, Kensington, WA 6151, Australia
[5]School of Engineering, University of Western Australia, Crawley, WA 6009, Australia
[6]Department of Civil and Environmental Engineering, Massachusetts Institute of Technology, Cambridge, MA, USA

**Correspondence:** Klaus Regenauer-Lieb (regenau@gmail.com)

**Abstract.** We propose a multiscale approach for coupling multiphysics processes across the scales. The physics is based on discrete phenomena, triggered by local Thermo-Hydro-Mechano-Chemical (THMC) instabilities, that cause cross-diffusion (quasi-soliton) acceleration waves. These waves nucleate when the overall stress field is incompatible with accelerations from local feedbacks of generalized THMC thermodynamic forces that trigger generalized thermodynamic fluxes of another kind. Cross-diffusion terms in the $4 \times 4$ THMC diffusion matrix are shown to lead to multiple diffusional $P$- and $S$-wave equations as coupled THMC solutions. Uncertainties in the location of meso-scale material instabilities are captured by a wave-scale correlation of probability amplitudes. Cross-diffusional waves have unusual dispersion patterns and, although they assume a solitary state, do not behave like solitons but show complex interactions when they collide. Their characteristic wavenumber and constant speed define mesoscopic internal material time-space relations entirely defined by the coefficients of the coupled THMC reaction-cross-diffusion equations. A companion manuscript proposes an application of the theory to earthquakes showing that excitation waves triggered by local reactions can, through an extreme effect of a cross-diffusional, wave operator, lead to an energy cascade connecting large and small scales and cause solid-state turbulence.

## 1 Introduction

The theory presented in this paper grew out of the conference series dedicated to understanding Coupled Thermo-Hydro-Mechanical-Chemical (THMC) in Geosystems (GEOPROC). The 7th international event was held in 2019 in Utrecht and focussed on earthquake and faulting mechanics (this special volume). Integration of mechanical, hydrodynamical, thermal, and chemical processes covers, however, a much wider field from the pore to plate-tectonic scale for a wide range of natural and engineering problems in geological system discussed in focus topics on earlier GEOPROC conferences. These problems include nuclear waste disposal, coal seam gas, enhanced oil and gas recovery, geothermal energy, mineral deposits, tailing dam collapse, landslide and many others. The individual problems may have their own characteristics. However, the common scientific issue of multiscale feedback of THMC processes remains the same.

The GEOPROC theme seeks to foster the urgently needed growth of experimental, numerical, and theoretical studies on multiphysics (THMC) and multiscale framework studies in Earth Sciences. The current practice is to still use engineering solutions based on empirical material laws to address specific natural and engineering problems in geological systems and energy production in geothermal energy, nuclear waste disposal, reservoir engineering for oil and gas, the formation of mineral deposits, induced seismicity, natural hazards, and $CO_2$ sequestration and utilization. These empirical engineering approaches are often inadequate, as indicated, for example, in the failure to avoid the 5.5 magnitude earthquake in Pohang Korea in November 2017, which was anthropogenically induced by high-pressure hydraulic injection during the previous two years (Grigoli et al., 2018).

Part of the reasons for lack of a wider adoption of coupled THMC approaches in the community is a lack of a theoretical basis on which to assess the rich solution space that arises from a coupling of the four (THMC) partial differential reaction diffusion equations. While parallel numerical tools for modelling fully coupled non-linear systems of THMC equations have become available through pioneering work in nuclear engineering (Gaston et al., 2009; Permann et al., 2020), the corresponding theory has not progressed as far. The application of the powerful nuclear engineering modelling tool has been successfully transferred to geosciences and applied to geodynamic modelling (Jacquey and Cacace, 2020a, b) and the modelling of the Non-Volcanic Tremor and Slip (NVTS) events in the circum-Pacific subduction zones (Poulet et al., 2014b) as well as applied to geological faulting problems (Poulet et al., 2014a). However, a sound theoretical description and interpretation of the local processes resulting in the interesting macroscopic phenomena has been lacking. The companion article (Regenauer-Lieb et al., 2020) aims at providing a detailed, step-by-step, explanation of the new theory used to rectify this shortcoming preceded by a short introduction into the theory of excitable waves triggered by THMC reaction terms.

Before discussing a possible application of the new theory to the processes of earthquakes and faulting in our companion article (Regenauer-Lieb et al., 2020), here we present a transdisciplinary approach bridging the gap between observations of instabilities from the molecular scale to the very large scale. The theory in this paper is written using approaches familiar to the theoretical and applied mechanics community. The original work is based on the 1960's work (Hill, 1962) building the foundation of theoretical approaches to localisation criteria, via the so-called acoustic tensor criterion, widely used in the engineering community (Rudnicki and Rice, 1975). The approach focusses on standing-wave quasi-static solutions based on vanishing speeds of acceleration waves which, without consideration of additional length scales, leads to infinite values of variables on the localisation bands such as infinite strain-rate in shear or infinite pressure (Veveakis and Regenauer-Lieb, 2015) for volumetric localisation bands. Surprisingly, little effort has been made to explore the rich wave field of the corresponding travelling-wave solutions, probably because dynamic events are only of academic interest to the engineering plasticity community that focusses mainly on developing safety standards as well as limit analysis and design. A notable exception is the work of Benallal and Bigoni (2004) who found that under dynamic conditions, unbounded growth of perturbations can be found in the short-wavelength regime with divergence growth.

While applied mathematical solutions exist, the preference in geosciences is to address the problem of unbounded growth by explicit consideration of additional physics. A specific case was shown where the infinite response can be captured by postulating a carefully chosen chemical reaction Alevizos et al. (2017). A recent contribution has introduced a complex multiphysics

approach to compaction band formation (Jacquey et al., 2021) by adding a diffusion mechanism to the carefully chosen reaction term. The solution space was explored numerically showing standing-waves that can interfer with a propagating wave, and also lead to a pattern with spatial periodicity. Both approaches solve the ill-posed problem for some cases but a general solution that uses the physics of internal processes to regularise the problem was lacking. This calls for an extension to the theoretical work of Hill (1962) which is presented here.

The dynamic field is of special interest to the researcher in the area of earthquake and faulting instabilities. The state of the art in this field is defined by the influential experimental work of Dieterich (1979) including the work on the application of the rate and state variable friction approach to earthquakes (Tse and Rice, 1986). The approach based on these laboratory-derived constitutive equations has reached a mature stage, and no attempt is made here to compare the rich field of findings with the present theory. We approach the problem from an entirely different angle through theoretical investigation of the mathematical solutions of the system of coupled partial differential THMC equations that deliver wave solutions with short-wavelength instabilities. In the course of developing the new approach, we describe wave physics phenomena that have previously not been reported in the solid earth community but are well known in a range of different fields from quantum systems to ocean waves (Zakharov et al., 2004). It is fair to say that the theory is rather in its infancy state, and special care needs to be taken before considering a direct application to the aforementioned systems. The first part therefore presents the theoretical derivation, and the second part delves into possible applications and proposed experiments to test the applicability of the theory.

In this paper, we introduce the classical approach of acceleration waves in plasticity theory to the seismology community by starting with the Helmholtz decomposition of the seismic wave equation into P- and S- waves (see section 3.1). We show how plasticity theory can be integrated into the equations via Hill's acceleration waves. This approach leads directly to the unbounded short-wavelength growth described by Benallal and Bigoni (2004) which cannot be solved without further assumptions. The innovation proposed in the two manuscripts is to appeal to the multiscale nature of the THMC coupled problem. We regularise the problem by embedding an open system thermodynamic multi-scale theory with unbounded solutions into a closed system macro-scale approach that describes the emergence of a standing-wave solutions.

There are two opposite starting points for the derivation of the approach. Here, we investigate the meso-scale from the conventional mechanical quasi-steady state (infinite time scale) solution of the macro-scale. Although the present manuscript uses the macro-scale perspective, i.e. the classical mechanical viewpoint for the investigation of the physics of acceleration waves (Hill, 1962), in variance to the classical approach we acknowledge, however, that both meso-scale processes and the macro-scale behaviour have an effect on each other. The companion article (Regenauer-Lieb et al., 2020) describes the meso-scale view which is the classical viewpoint of a chemist. Both viewpoints are objective descriptions of the coupled THMC problem and should deliver the same outcome.

In order to define the separation between the meso- and macro-scale of a THMC coupled problem, we propose that the scale for each of the THMC-processes is defined by its own characteristic diffusion time/length scales (Regenauer-Lieb et al., 2013b). The THMC diffusion length scale is thereby related to the time scale of a considered THMC process as defined by the proportionality to the square root of the diffusivity multiplied by the process time. For simple problems progress can be made by studying thermodynamic equilibrium states in isolated closed systems. Likewise closed, coupled, far-from-equilibrium THMC

systems that feature irreversible behaviours can be modelled by a thermomechanics approach (Collins and Houlsby, 1997), also called a thermodynamics with internal variables approach (Maugin and Muschik, 1999) or a hyperplastic approach (Houlsby and Puzrin, 2007). This theory is, however, only applicable to faults that have reached a thermal steady state as implied by a standing-wave solution of acceleration waves. This approach prevents modelling of dynamic phenomena. Modelling of earthquake and faulting is hence one of the most difficult topics to address using a self-consistent thermodynamic approach.

A particular challenge for deriving dynamic THMC coupled wave solutions is the discrete nature of the cascade of steady state solutions defined by the standing-wave solutions of thermomechanics which leads to a discrete material behaviour as discussed in the next section. Standard probability theory is therefore not suitable as this assumes a continuum of wave functions (Cohen, 1988). In order to solve this issue we use a transfer of knowledge from classical quantum mechanics to characterise any system at a larger scale. The information on multiple internal material time/length scale processes disperses each at characteristic velocities in the form of acceleration waves.

The nucleation mechanism of these waves relies on the meso-scale open system behaviour where the overall macro-scale thermodynamic forces can become incompatible with accelerations from local thermodynamic fluxes. These incompatibilities radiate wave energy away from its source in the form of "cross-diffusion" waves. The emergence of cross-diffusion waves can be perhaps best understood from a chemical viewpoint (Regenauer-Lieb et al., 2020) where propagating chemical waves have been studied in detail (Vanag and Epstein, 2009). In chemical systems, cross-diffusion is defined as the phenomenon in which a gradient in the concentration of one species induces a flux of another chemical species. In the present context thermodynamic forces and fluxes are generalized THMC fluxes defined in Table 1. Before discussing cross-scale coupling of thermodynamic forces and fluxes in sections (3.3) following, it is useful to briefly review insights into the formation of discrete dissipative structures.

## 2  Dissipative structures

The concept was introduced first in chemical and biological systems where morphogenic patterns (Turing, 1952) were identified as solutions to the underlying reaction-diffusion equations (see Figure 1). These discrete patterns were later on named Turing patterns. A review of Turing patterns in nature can be found in Ball (2012).

We propose here a generalised approach to cross-diffusion that is known in bio-physics as taxis (Heilmann et al., 2018). For example, the pufferfish and the siltstone in Figure 1 show similar patterns caused by a fundamental mechanism known as taxis. This is a process that forces components of a pattern to organise as an ensemble in reaction to changes in the environment, and so to move towards, or away, from a perturbation. The process eventually leads to the formation of a new energetically stable pattern. The fundamental pattern-forming taxis mechanism can be caused by adhesive forces (hapto-taxis), hydrodynamic (gyro-taxis), gravitational (gravi-taxis), light intensities (photo-taxis), or chemical driving forces (chemo-taxis), as shown in Figure 1. Non-biological patterns are generally formed by THMC reactions, which can also include electrical and biological drivers. In mathematical biology, taxis models are used to understand and quantify a variety of complex problems, ranging

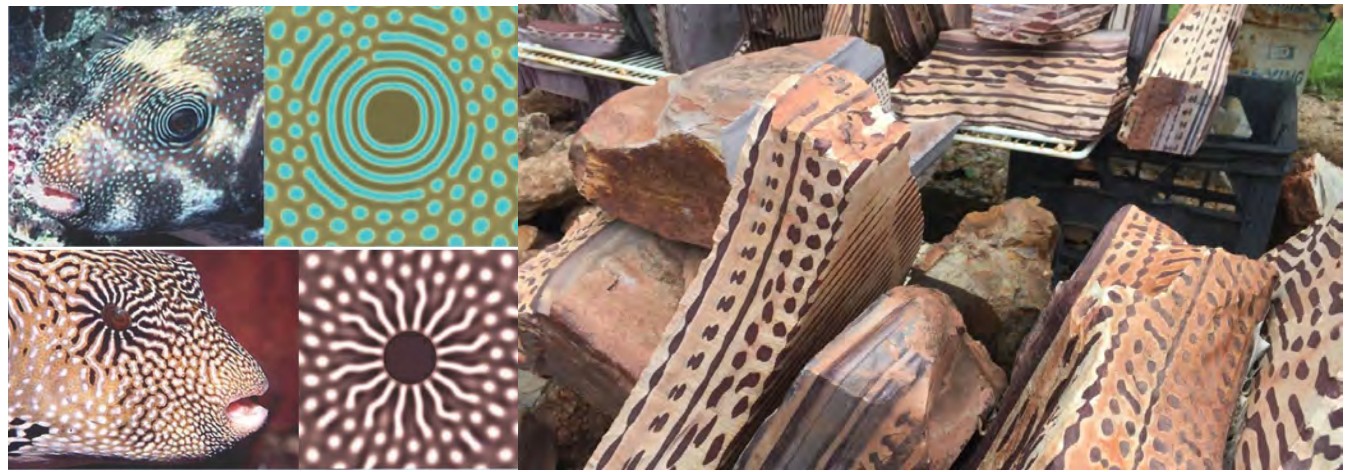

**Figure 1.** Dissipative patterns in chemical and biological systems: Simulated and real patterns on Puffer Fish (Sanderson et al., 2006) and strangely patterned siltstone (Zebra Rock, Ranford formation in the Kununurra district in the East Kimberley region of Western Australia). Both pattern formations can be modelled by a reaction-diffusion-advection type instability due to chemo-taxis.

from nerve pulse responses and spreading of diseases (Zemskov et al., 2017) to predicting the spatio-temporal patterns of
predator-prey systems.

However, except for the seminal early work by Ortoleva and co-workers (Dewers and Ortoleva, 1990; Ortoleva, 1993, 1994) developments of taxis models in Earth and Material Sciences have lagged. A review of the progress made in this field as well as a specific case study of rhythmic banding in marls can be found in the recent work of L'Heureux (L'Heureux, 2018, 2013). The present work develops the key ideas into a geomechanical perspective building on an initial approach proposed for hydro-
mechanical coupling (Hu et al., 2020; Alevizos et al., 2017; Regenauer-Lieb et al., 2016; Veveakis and Regenauer-Lieb, 2015; Regenauer-Lieb et al., 2013a).

By analogy to the mathematically similar biological and chemical systems, we propose here that earthquake instabilities are preceded (and followed in the post-seismic stage) by propagating THMC dissipative waves which could enable new detection methods if they can be resolved by sensors. We will discuss such possible precursor phenomena for earthquakes in the compan-
ion article (Regenauer-Lieb et al., 2020). In chemical systems, propagating waves stemming from reaction-diffusion processes are very well documented and the hypothesis here is that this applies generally to all THMC coupled processes. A review and update of the formulation for chemical systems can be found in Vanag and Epstein (2009). While chemical oscillations thus appear to be well understood, the phenomenon of an oscillatory response is less well established in other THMC systems. However, these systems show the same transitions from a simple continuum response to a highly localized state. The existence
of a discrete, particle-like nature has also been discovered in fluids when they are driven far from equilibrium. If driven far from equilibrium by surface forces, fluids clearly show (see Fig. 2) a highly dissipative, sharp transition from a continuum state to one of a highly localized, propagating, particle-like state (Lioubashevski et al., 1996).

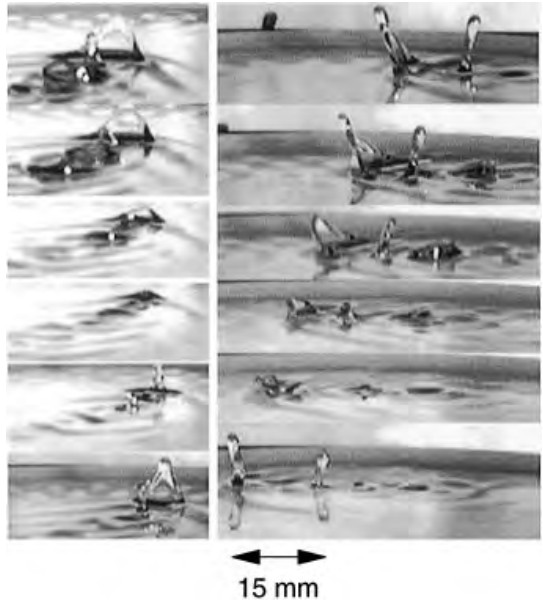

15 mm

**Figure 2.** Dissipative patterns in fluid systems: Water molecules exhibit a discrete quantum-like solitary state when forced by a mechanical shaker at a critical condition (here 41 Hz). Periodic finger-like solitary states travel from right to left at a constant velocity. Each snapshot shows 20 ms intervals. Unlike classical solitons their appearance is particle-like. They can pass through each other with a slight loss of amplitude, or 'collide' to create a new state whose direction of propagation is at an angle to that of the original states or disintegrate upon collision (image from Lioubashevski et al. (1996)) with copyright permission for the American Physical Society article "Dissipative Solitary States in Driven Surface Waves" identification number RNP/20/OCT/032299.

For the case of deforming geomaterials, a theory for localization phenomena, characterized by a sudden transition from continuum deformation behaviour to a highly localized state, is well established (Rudnicki and Rice, 1975). Fig. 3 shows a periodic set of localised deformation structures formed as a result of a compressive tectonic regime. Similar standing-wave like features are encountered in many geological systems (L'Heureux, 2013; Ball, 2012). However, direct experimental evidence for precursory transient travelling solitary states is largely unknown and has only been shown recently based on mathematical considerations (Hu et al., 2020). In the supplementary material of the companion article (Regenauer-Lieb et al., 2020) we will discuss possible experimental tests of the precursor phenomena. The lack of experimental evidence can be explained by the challenging task of dealing with the large length scale of the geomechanical phenomena and the long-time scales of observation required to mimic natural processes in the laboratory (Paterson, 2001).

The dynamics of the formation of these mechanical dissipative patterns can therefore only be investigated using analogue materials in the laboratory. Analogue experiments have been performed in a granular, brittle- matter compressed uniaxially (Guillard et al., 2015; Einav and Guillard, 2018). A propagating compaction wave phenomenon has been observed. Acoustic bursts have been registered when the waves interact with interfaces leading to the conversion of their kinetic energy into acoustic emissions. The phenomenon has been compared to ice-quakes in ice sheets (Einav and Guillard, 2018). While these

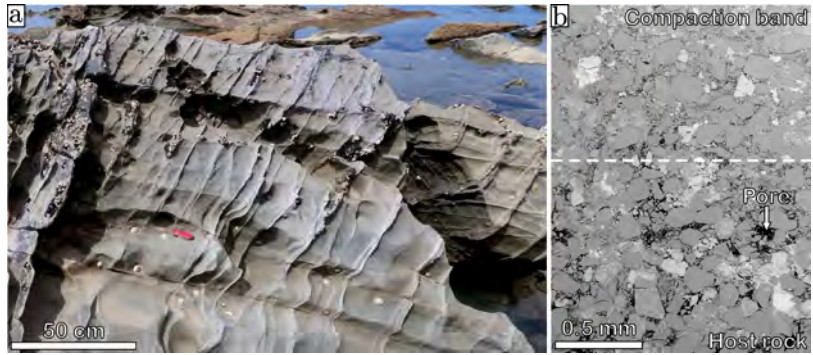

**Figure 3.** a) Photograph of three generations of shear-enhanced compaction bands and pure compaction bands in silt- and sandstones of the Miocene Whakataki Formation, South of Castlepoint, North Island, New Zealand. These deformation bands formed during the slow compression of the Hikurangi subduction wedge (see Elphick et al. (2021) for details). The positive relief of the metre-long bands is caused by the significant porosity reduction relative to the host rock, which renders them less susceptible to erosion. The dominant microphysical deformation mechanism for inelastic volume loss is grain crushing (see panel (b)). The bands possibly formed by hydro-(chemo)-mechanical coupling, which can explain the relatively short (diffusive) length scales (Hu et al., 2020). The regular, constant spacing may be indicative of the standing-wave phenomenon (Regenauer-Lieb et al., 2013a; Veveakis and Regenauer-Lieb, 2015). (b) Scanning-electron micrograph, recorded with the backscattered electron detector, of a compaction band (region above the white dashed line) and its host rock in fine-grained sandstone of the Whakataki Formation. Pores appear black (the white arrow marks an example). The compaction band displays a marked reduction in porosity and contains a much larger proportion of crushed grains than the host rock.

experiments allow some insight into the precursor phenomena of stationary compaction bands, the experiments themselves never reached the stationary mode. This aspect will be discussed in more detail in the companion article (Regenauer-Lieb et al., 2020) where experiments that are very close to the stationary mode are also introduced (Barraclough et al., 2017). The stationary mode allows development of a robust *thermomechanics* theory which offers a modular thermodynamically self-consistent approach for modeling earth instabilities (Jacquey and Regenauer-Lieb, 2020).

Experiments with highly porous carbonates have been performed (Chen et al., 2020). These produced stationary and non-stationary compaction bands under uniaxial loading. Unfortunately, these experiments have the opposite problem in that the exact analysis of the dynamic evolution did not reach sufficient resolution in space and time to convincingly detect the wave phenomenon described in the granular brittle matter. We, therefore, explore in this contribution theoretical predictions of the dynamic wave propagation based on an extension of the *thermomechanics* approach for wave propagation in dissipative materials (Coleman et al., 1965).

We propose here that the dissipative wave phenomenon is universal for THMC reaction-diffusion systems that are driven far from equilibrium. The approach allows an interpretation of observations in nature and the laboratory in terms of propagating particle-like states which emerge as stationary Turing patterns for long-timescale standing-wave solution of a THMC cross-diffusion formulation. In order to recover the dissipative wave equations, we present in the following the standard constitutive

assumptions for any generic thermodynamic fluid or solid mechanical system and describe how the physics of THMC feedbacks can be implemented to resolve the phenomenon of propagating dissipative waves in these systems.

## 3 Wave equations

### 3.1 Constitutive assumptions

The fundamental equation of motion is:

$$\nabla \cdot \boldsymbol{\sigma} + \mathbf{f} = \rho \mathbf{a}, \tag{1}$$

where $\boldsymbol{\sigma} = \sigma_{ij}$ is the Cauchy stress tensor, $\rho$ the density, $\mathbf{f}$ is a body force (e.g. gravity) and $\mathbf{a}$ the acceleration. This equation does not stipulate a constitutive law but with constitutive assumptions it becomes the master equation for the theory of elastic waves, fluid mechanics and continuum mechanics. In elasticity, wave equations directly result from the equation of motion defining the wave characteristics by using the Helmholtz decomposition, in terms of shear ($S$-wave) and compressional ($P$-wave) wave velocities which is a convenient description for the purpose of this paper.

For an isotropic elastic medium, for instance, accelerations in Eq. (1) are only allowing elastic displacements described by $\mathbf{u}$. In this case the material can be characterized by just two velocities: the elastic $P$-wave velocity $v_p$ and the elastic $S$-wave velocity $v_s$, and we obtain from Eq. (1) the elastic-wave equation:

$$\frac{\partial^2 \mathbf{u}}{\partial t^2} = \underbrace{v_p^2 \nabla(\nabla \cdot \mathbf{u})}_{P \text{ wave}} - \underbrace{v_s^2 \nabla \times (\nabla \times \mathbf{u})}_{S \text{ wave}}. \tag{2}$$

Similarly, by allowing the material to deform in a viscous manner, acceleration can be monitored by a local change in velocity $\mathbf{v}$, and the Helmholtz decomposition identifies a scalar $P$-wave and a vectorial $S$-wave potential field. The material constants are the dynamic shear $\eta$ and bulk $\zeta$ viscosities to obtain the generalized Navier-Stokes equation:

$$\rho \left( \frac{\partial \mathbf{v}}{\partial t} + \mathbf{v} \cdot \nabla \mathbf{v} \right) = -\nabla p + 2\nabla^2(\eta \dot{\boldsymbol{\epsilon}}') + \nabla(\zeta(\nabla \cdot \mathbf{v})) + \mathbf{f} \tag{3}$$

where

$$\dot{\boldsymbol{\epsilon}}' = \frac{1}{2} \left( \nabla \mathbf{v} + (\nabla \mathbf{v})^T \right) - \dot{\boldsymbol{\epsilon}}_0$$

is the deviatoric viscous strain-rate, with

$$\dot{\boldsymbol{\epsilon}}_0 = \frac{1}{3}(\nabla \cdot \mathbf{v})\mathbf{I}$$

where $\mathbf{I}$ is the identity matrix.

We emphasize here, that although the Helmholtz decomposition can be performed in a similar way to derive volumetric and shear moduli that describe dissipative material behaviour, their response to infinitesimal perturbation is generally to dampen propagating elastic waves. One could, therefore, come to the erroneous conclusion that their contribution to precursory wave phenomena to macroscopic failure is an overall suppression of instabilities.

Coleman and Gurtin (1965) have shown that this conclusion is wrong using the concept of materials with fading memory conceptualized by rational thermodynamics. We are using a simpler approach and are introducing fading memory through THMC dissipation processes based on the non-equilibrium thermodynamics approach of deGroot (1962). Therein, non-equilibrium conditions are seen as a concatenation of thermostatic equilibrium processes. We, therefore, can use the local equilibrium definition of the pressure as $p = -\frac{\partial U}{\partial V}$, where $U$ is the internal energy and $V$ the volume and explore the emergent dynamics through investigating the stability of small perturbations from individual equilibrium states. In the present context pressure is therefore defined as $p = \frac{1}{3}tr(\boldsymbol{\sigma})$ and is negative for compression.

For the elasto-visco-plastic case, we have the equivalent fourth-order elastic-viscoplastic stiffness tensor $\mathbf{C}$ characterizing material stiffness and the corresponding elasto-viscoplastic bulk viscosity $\zeta$ to give

$$\rho \left( \frac{\partial \mathbf{v}}{\partial t} + \mathbf{v} \cdot \nabla \mathbf{v} \right) = -\nabla p + 2\nabla(\boldsymbol{C}\dot{\boldsymbol{\epsilon}}') + 3\nabla(\zeta\dot{\epsilon}_0) + \mathbf{f}, \tag{4}$$

in this case $\dot{\boldsymbol{\epsilon}}'$ denotes the deviatoric strain-rate which in the purely elastic case before yield is $\dot{\boldsymbol{\epsilon}}' = \dot{\boldsymbol{\epsilon}}'_e$ becoming post-yield the elasto-viscoplastic strain-rate defined by $\dot{\boldsymbol{\epsilon}}' = \dot{\boldsymbol{\epsilon}}'_e + \dot{\boldsymbol{\epsilon}}'_{vp}$ where the subscripts $e$ and $vp$ refer to the elastic and viscoplastic components. The same approach is used for decomposing the equivalent elasto-viscoplastic volumetric strain-rate $\dot{\epsilon}_0$. While the emergence of elastic $P-$ and $S-$ waves for any infinitesimal disturbance is a well-known physics phenomenon, dissipative processes are commonly known to dampen elastic waves. Long-range wave propagation in dissipative materials was therefore contested for a long time (Coleman and Gurtin, 1965). Ideal elastic waves without damping propagate without loss of energy as they are based on the conservation of energy. However, the dissipative chemical and biological diffusion waves are known to propagate as autonomous wave sources, spontaneous oscillations, and quasi-stochastic waves which are synchronized over the entire space to form dissipative structures (Vasil'ev, 1979).

They are an entirely different class of waves as they are based on dissipation in active kinetic systems in contrast to waves in conservative systems. For simplicity, we only discuss the slow visco-plastic wave phenomenon allowing the investigation of conservative and dissipative waves as different processes. For decoupling elastic and dissipative waves, we need to assume large differences in the propagation speed of the waves. This is done by assuming Maxwellian rheology, implying a separation of elastic and visco-plastic wave time-scales in the context of an additive strain-rate decomposition of Eq. (2) and (4). To recover dissipative waves from the above discussed Navier-Stokes equation, modified for the inclusion of elastoplastic behaviour, we introduce local thermodynamic THMC feedback processes that change the instantaneous fourth-order elastic-visco-plastic stiffness tensor $\mathbf{C}$.

## 3.2 Acceleration waves and classical theories of localisation

The simplest implementation of the non-equilibrium approach of deGroot (1962) is the theory of internal variable thermodynamics which unifies the kinetic description of chemical reaction-diffusion processes and the above described elastic-viscoplastic formulation (Maugin and Muschik, 1999). Perturbations to the local equilibrium assumption of the non-equilibrium thermodynamic theory of internal variables can lead to conditions of violation of smoothness on surfaces in a body, where one or more internal variables from the lower scale suffer jump discontinuities, owing to locally reaching a critical dissipation. In

a classical thermodynamic sense, this can be viewed as suddenly switching on a micro-engine somewhere in the system that disturbs the overall stress field. This is the physical reason for the formation of acceleration fronts, where the diffusive length-scales are linked to the convective velocity of the step function on dissipation waves. While fluid- and solid-wave phenomena occur when the above equation includes inertial forces, so-called 'acceleration waves'(Hill, 1962), caused by local surfaces of acceleration (see Fig. 4), can also occur in the creeping flow limit when no acceleration due to a (gravity) potential is present and $\mathbf{f} = 0$. These acceleration waves are defined as geometric surfaces (here assumed to be plane waves) that move relative to the material.

Acceleration waves can be described in two ways. One can use two coordinate systems, one for the reference state and one for the current state. A more elegant way is to consider convective coordinate systems by formulating the constitutive law in terms of stress-rate. For this we consider the space derivative normal to the moving wavefront (see Figure 4) indicated by $\frac{\partial}{\partial s}$. Waves are travelling concerning a background Lagrangian moving material reference frame.

Considering the traction (load per unit area) in the direction normal to the wavefront as $\mathbf{F}$ and choosing the magnitude of the velocity of the moving wavefront as $c$, the jump condition indicated by the square Iverson brackets can be advected along $c$. This leads to Hadamard's jump condition where the true traction rate along the advected coordinates is:

$$\left[\dot{\mathbf{F}}\right] = -c\left[\frac{\partial \mathbf{F}}{\partial s}\right]. \tag{5}$$

Hadamard's jump condition applies to all internal variables and the acceleration across the wavefront is constrained by

$$[\rho\dot{\mathbf{v}}] = \left[\frac{\partial \mathbf{F}}{\partial s}\right]. \tag{6}$$

Combining Eq. (5) and (6) we obtain

$$[\dot{\mathbf{F}}] + c[\rho\dot{\mathbf{v}}] = 0. \tag{7}$$

Substituting $\dot{\mathbf{v}} = -c\frac{\partial \mathbf{v}}{\partial s}$ into Eq. (7) we obtain

$$[\dot{\mathbf{F}}] = c^2\left[\rho\frac{\partial \mathbf{v}}{\partial s}\right]. \tag{8}$$

Hill's formulation of acceleration waves in Eq. (8) expresses the energetics of the acceleration waves by the square of the material velocity $c$ times the mass of the characteristic segment defined by $\frac{\partial \mathbf{v}}{\partial s}$. This provides a simple formulation where the energetics of the material is solely described by Eq. (1) and the meso-scale mass exchange rates on acceleration waves by Eq. (8). The material velocity $c$, being the velocity of acceleration waves becomes a material constant for the propagation of acceleration waves. Acceleration waves form the basis of localization criteria in plasticity theory. The criterion for instability is derived from the equivalent theory in elastodynamics where for an elastoplastic body the acoustic tensor $\mathbf{\Gamma}$ is defined by

$$\mathbf{\Gamma} = \mathbf{n} \cdot \mathbf{C} \cdot \mathbf{n} \tag{9}$$

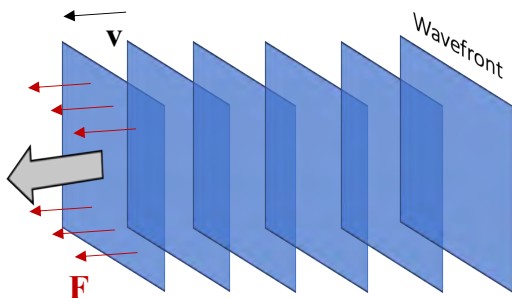

**Figure 4.** Acceleration waves can originate at a body surface when the existing internal stress gradient is dynamically incompatible with accelerations imposed on particles of the surface. A propagating plane-wave front is shown here for reference, but a plane-wave is not a necessary restriction. Across these surfaces particle accelerations and spatial gradients of velocity are momentarily discontinuous while the velocity itself is continuous.

In elastodynamics, the eigenvalues of $\mathbf{\Gamma}$ divided by the mass density represent the square of the elastic wave propagation speed in the direction of the unit normal vector $\mathbf{n}$. In elasto-plasticity, the equivalent dynamic stability criterion is defined by Eq. (8) which in terms of acoustic tensor implies that

$$\mathbf{\Gamma}\frac{\partial \mathbf{v}}{\partial s} = \rho c^2 \frac{\partial \mathbf{v}}{\partial s} \tag{10}$$

Dynamic system stability can be evaluated through assessing the eigenvalues of the acoustic tensor thus determining the speed of the acceleration waves which must be real and defined by the square root of the instantaneous modulus $\mathbf{C}$ divided by the instantaneous density $\rho$ (Coleman and Gurtin, 1965). Mathematically, Eq. (2 - 4) can be represented by the addition of two functions, a scalar field and the curl of a vector field. The scalar field without curl or rotation identifies dissipative compressional $P$-waves and the curl features zero divergence and corresponds to isochoric sinistral and dextral dissipative shear $S$-waves.

These waves are interpreted as stationary (standing) waves when the determinant of the acoustic tensor and consequently the wave speed is zero,

$$det(\mathbf{\Gamma}) = 0 \tag{11}$$

which is the standard condition for localisation in plasticity theory (Vardoulakis and Sulem, 1995). Accordingly, the formation of localized shear-bands out of homogeneous plastic flow is assumed, when the velocity of the wavefront vanishes. Hill (1962) was discussing shear acceleration waves in an ideal linear, time-independent elasto-plastic material where two families of characteristics (dextral and sinistral slip lines) feature a jump in strain-rate at the wavefront accompanied by one in stress-rate (but not in stress). This in turn implies a related jump in stress gradient. Later work extended the theory to formulate

accelerations waves as the basis of modern criteria for localization in plastic media (Rudnicki and Rice, 1975; Rice, 1976).
In those theories the possibility of volumetric acceleration waves was, however, neglected and volumetric deformation was parameterized by an empirical dilatancy angle. Another shortcoming of the localization criterion for the application to THMC instabilities is that it is not directly applicable to the rate-dependent elasto-viscoplastic case. The inclusion of rate effects implies a positive wave speed different from zero (Duszek-Perzyna and Perzyna, 1996). The *thermomechanics* approach (Jacquey and Regenauer-Lieb, 2020) allows incorporation of a quasi-static wave speed related to the internal variable that quantifies the rate-dependence.

However, to date, no generally accepted localization criteria for the transition from a dynamic to quasistatic rate-dependent solution of Eq. (4) exists, although stationary and wave-like propagating localization phenomena for rate-sensitive materials (Barraclough et al., 2017) are observed in the laboratory and nature. The method of choice to date is to use all field equations and perform a numerical stability analysis. A discussion on an extension to the above-discussed criterion has been presented recently (Pisanò and Prisco, 2016) and energy-based criteria that successfully model the adiabatic limit have been repeatedly revisited many times over the past 30 years (Paesold et al., 2016). The present approach provides an alternative path to systematically analyse the full system of field equations.

### 3.3 THMC acceleration waves

We assume creeping flow in Eq. (4), and there is, therefore, no effect of inertia ($\mathbf{F} = 0$) but there still can be effects of "gravitational acceleration", i.e. creeping flow in a gravity field. We will show that the meso-scale formalism identifies an alternate internal force density from within the considered material volume stemming from a local thermodynamic THMC force (e.g. $\nabla p$). This internal force integrates over the accelerations $\mathbf{a_M}$ of the micro-processes inside the continuum element by multiplying them with the average volume density. These accelerations stem from dissipative mechanisms (e.g. volume changes by phase transitions, fracture, etc.) inside the representative volume element (Hu et al., 2020). For critical conditions, they can cause acceleration waves propagating as creeping waves. Hadamard's jump conditions need to be extended for internal THMC variables $\mu$ such as temperature, porosity, permeability, viscosity, etc...

Hadamard's jump conditions state that if time derivatives ($\dot{\mathbf{F}}$, $\dot{\mathbf{v}}$, $\dot{\mu}$) and gradients $\nabla \mathbf{F}, \nabla \mathbf{v}, \nabla \mu$ have jump discontinuities across the wavefront then $\mathbf{F}, \mathbf{v}$ and $\mu$ are continuous functions of space. The compatibility condition relates jumps in rates of change of internal variables to jumps in gradients for all internal variables $\mu$ (Duszek-Perzyna and Perzyna, 1996) and implies that the jump in the gradient of pressure inside the acceleration wave is constrained by

$$[\nabla p] = -\frac{1}{c}[\dot{p}]. \tag{12}$$

Acceleration waves consider a step function (Eq. 5) where the stress-rate is discontinuous along the surface. The stress is, however, continuous across the wavefront and the stress self-diffusion coefficient is also constant outside of the wave. Therefore, for modelling acceleration waves in a homogeneous material we can simplify Eq. (1) further and assume constant bulk and shear viscosity outside the wave and assume continuity of stress across the acceleration wave. Noting that the traction in the direction of the normal vector $\mathbf{n}$ on the acceleration wavefront is $\mathbf{F} = \mathbf{n} \cdot \boldsymbol{\sigma}$. It follows from Eq. (7) that the jump in stress-rate

on the acceleration wave is (Duszek-Perzyna and Perzyna, 1996):

$$\mathbf{n} \cdot [\dot{\boldsymbol{\sigma}}] = -c[\rho \dot{\mathbf{v}}] \tag{13}$$

Substituting the stress-rate for the acceleration $\dot{\mathbf{v}}$ from Eq. (13) and the pressure rate for the gradient of pressure from Eq. (12) and inserting the jump condition into Eq. (4) it follows that

$$\frac{1}{c}[\mathbf{n} \cdot [\dot{\boldsymbol{\sigma}}]] = -\frac{1}{c}[\dot{p}] - [\boldsymbol{C}\nabla(\dot{\boldsymbol{\epsilon}}')] - [\zeta\nabla\dot{\boldsymbol{\epsilon}_0}] \tag{14}$$

If we define the magnitude of the wave speed in the normal reference system as $w = \mathbf{w} \cdot \mathbf{n}$, then $c = w - \mathbf{v} \cdot \mathbf{n}$ is the local particle velocity of THMC accelerations in the acceleration wave relative to the normal material velocity.

Eq. (14) allows us to draw some important conclusions for acceleration waves. (1) The first term on the right shows that the pressure rate divided by the wave velocity or the equivalent gradient of pressure plays an important role in acceleration waves. (2) The second term on the right implies that gradients of deviatoric strain-rates are related to rate changes of the stiffness tensor as implied by the jump condition of the internal variable inside the propagating wave. Recall that the jump condition (Eq. 5 or 12) advects jumps in gradients of the internal variable around the propagating wavefront through a jump in the rate of change of the variable. (3) The last term implies that the same is true for the volumetric strain-rates and the rate of change of bulk viscosity.

## 4 Multiscale cross-diffusion model

So far we have only discussed the mechanical reaction-diffusion equation, where the shear and bulk viscosities control the diffusion of stress. For the multiphysics implementation, it is convenient to think of diffusion of momentum and use the momentum diffusivity (kinematic viscosity) instead of the dynamic viscosity. We, therefore, denominate $\zeta_M$ as the volumetric diffusion coefficient of pressure (kinematic viscosity). In the following, we first formulate the reaction-diffusion equation in a classical way. That is to say that meso-scale cross-diffusion effects are neglected. We identify THMC-Turing patterns as multiscale energy eigenstates of the reaction-diffusion equations thus characterizing Prigogine's dissipative structures if they emerge.

In these formulations, the viscous (M) mechanical pressure diffusion equation finds its counterparts in the equivalent thermal (T) Fourier-, (H) Darcy- and (C) Fick's- diffusion laws where the diffusion coefficients are indicated by the associated THMC subscript. The corresponding reaction rates are the local hidden-variable reaction rate $R_T, R_H, R_M$ and $R_C$, respectively. It is common practice to ignore the meso-scale cross-diffusion kinetics introduced in the classical theories of localisation. We emphasize therefore the difference between large-scale reaction rates $R_i$ and meso-scale reaction rates $r_i$ of the multiscale theory which considers the important effect of cross-diffusion. The two rates are identical in the infinite time-scale limit as cross-diffusion can be eliminated adiabatically (Biktashev and Tsyganov, 2016).

In the adiabatic limit we obtain similar reaction-diffusion equations across a vast range of THMC diffusion length scales as tabulated in Table 1. The reaction rates most often stem from different micro-processes at lower scale inside the considered

**Table 1.** Generalized Thermodynamic Fluxes and Forces in a THMC coupled system (1-D)

| Type | Force | Flux | reaction-diffusion equations |
|------|-------|------|------------------------------|
| **T** | $F_T = \nabla T$ | $q_T = -\frac{DT}{Dt}$ | $\frac{DT}{Dt} = \zeta_T \nabla^2 T + R_T$ |
| **H** | $F_H = \nabla p_H$ | $q_H = -\frac{Dp_H}{Dt}$ | $-\frac{Dp_H}{Dt} = \zeta_H \nabla^2 p_H + \eta \dot{\epsilon}' - R_H$ |
| **M** | $F_M = \nabla p_M$ | $q_M = -\frac{Dp_M}{Dt}$ | $-\frac{Dp_M}{Dt} = \zeta_M \nabla^2 p_M + \nabla(\boldsymbol{C}\dot{\boldsymbol{\epsilon}}') - R_M$ |
| **C** | $F_C = \nabla C$ | $q_C = -\frac{DC}{Dt}$ | $\frac{DC}{Dt} = \zeta_C \nabla^2 C + R_C$ |

continuum element which introduces cross-scale diffusion fluxes as shown in the next section. In order to generalize the approach, we propose that all reaction-diffusion equations in Table 1 are strongly coupled. We construct a composite multiscale THMC diffusion wave operator $\hat{H}_{THMC}$ from the four reaction-diffusion equations in Table 1.

$$\hat{H}_{THMC} = -\zeta_i \sum_{i=1}^{N} \nabla^2, \tag{15}$$

where $\nabla^2 = \frac{\partial^2}{\partial x^2} + \frac{\partial^2}{\partial y^2} + \frac{\partial^2}{\partial z^2}$ and $i$ refers to the individual thermodynamic THMC process.

The wave operator $\hat{H}_{THMC}$ defines the asymptotic dynamic state of the coupled system of THMC reaction-diffusion equations by mapping excitations from reactions into a new waveform. It therefore selects the wave that we can expect from the discrete interactions at lower scale (e.g. atomic or molecular scale reactions in the chemical example). If we thus excite waves with the reaction term $R_i$, the approach can lead to unbounded instabilities because the statistical information from the lower scale interactions, such as a diffusional length scale that limits unbounded reactions on the lower scale, is missing. A particular strategy to include such information is to use nonlocal reaction-diffusion equations (Rubinstein and Sternberg, 1992). We therefore argue that the wave operator must include information from the lower scale. We propose that this information is contained in so-called "cross-diffusion coefficients" that are hidden at macro-scale in the reaction term $R_i$. This innovation regularises the ill-posed problem for all couplings and was first introduced for HM coupling in Hu et al. (2020). A very important aspect is that the inclusion of the cross-diffusion terms into the reaction-diffusion equation of Table 1 leads to a new form of soliton-like waves as clearly shown by Tsyganov et al. (2007) for a mathematically similar system of equations.

The discussion of these waves and their unusual properties will be the subject of the remainder of the manuscript. For introduction and completeness, however, the arguments for non-local reaction-diffusion equations with cross-diffusion terms (Hu et al., 2020) are repeated in the next section and generalised from poromechanics HM problems to all THMC processes.

For the following discussion, we also simplify further and neglect the deviatoric terms in Eq. 14 and retain only the scalar volumetric terms and reduce the equations to 1-D. This allows us to investigate the poorly known volumetric dissipative waves which must exist in addition to the dissipative shear waves discussed by Hill (1962). To introduce the meso-scale considerations we identify wave scale reactive source terms $r_T, r_H, r_M, r_C$ of the hidden variable rates $R_T, R_H, R_M, R_C$ as

the actual terms that trigger acceleration waves. These meso-scale source terms stem from a jump in the thermodynamic force (gradient of the variable) into a jump in the thermodynamic flux (rate of the variable, i.e. temperature, fluid, and solid pressure and concentration). The important volumetric coupling is overlooked in the classical localisation theory (Rudnicki and Rice, 1975). The wave-scale source term provides the convected pressure rate built up by internal accelerations. It relates to the local mass exchange processes according to Eq. 8.

## 5 Cross-diffusion as a multiscale theory for localisation

In the following we generalise the discussion of the meso-scale THMC mass exchange processes using mixture theory applied to HM coupling as presented in Hu et al. (2020). We show that the physics of cross-diffusion follows from a reactive source term at the macro-scale that requests a cross-diffusion term at the meso-scale for thermodynamic consistency. The full derivation is found in Hu et al. (2020). Here we summarize the main conclusion from the mixture theory analysis for convenience.

We consider two mass fractions $A$ and $B$ for mass exchange denoted by the $i^{th}$ and $j^{th}$ phase as an example. We identify $\dot{\xi}_i^{REV}$ as the large-scale Representative Elementary Volume (REV) for averaging of mass transfer rate from the phases $A$ to $B$ where $V_{REV}$, $V_A$, $V_B$ denote the REV volume and the volume of the $i^{th}$ and $j^{th}$ phase, respectively. $\dot{\xi}_i^{REV}$ defines the REV-scale averaging of the mass exchange rate between the phases where the REV-scale source term of mass is obtained from the other species:

$$\dot{\xi}_i^{REV} = \frac{1}{V_{REV}} \int\limits_{V_{REV}} \dot{\xi}_j^{local} dV_{REV}, \tag{16a}$$

$$\dot{\xi}_j^{REV} = \frac{1}{V_{REV}} \int\limits_{V_{REV}} \dot{\xi}_i^{local} dV_{REV}, \tag{16b}$$

where $\dot{\xi}_i^{local}$ and $\dot{\xi}_j^{local}$ denotes the mass exchange rate from the $A$ to $B$ phase and vice-versa. In the meso-scale formalism we need to consider information from the local scale processes in the THMC diffusion matrix and decompose the processes leading to the local mass production $\dot{\xi}_i^{local}$ and $\dot{\xi}_j^{local}$. In order to specify this further we define the global volume fraction of the $A$-phase as:

$$\phi = \frac{V_A}{V_{REV}} = 1 - \frac{V_B}{V_{REV}}, \tag{17}$$

Mass conservation at global scale for the phases $A$ and $B$ gives:

$$\frac{\partial[\rho_A V_A]}{\partial t} + \frac{\partial[\rho_A V_A v_A]}{\partial x} = \dot{\xi}_A V_{REV}, \tag{18a}$$

$$\frac{\partial[\rho_B V_B]}{\partial t} + \frac{\partial[\rho_B V_B v_B]}{\partial x} = \dot{\xi}_B V_{REV}. \tag{18b}$$

$\rho_A$ and $\rho_B$ identify the density of the respective phases, while $v_A$ and $v_B$ their velocities in the direction of $x$ while $\dot{\xi}_A$ and $\dot{\xi}_B$ represent the volume averaged mass generation in the REV. Following an approach presented in Hu et al. (2020) using the

generalised THMC mass exchange processes we arrive at:

$$\frac{\partial[\rho_A\phi]}{\partial t} + \underbrace{\frac{\partial[\rho_A\phi v_A]}{\partial x}}_{Self-diffusion} + \underbrace{\frac{1}{V_{REV}}\int_{V_{REV}}\frac{\partial[\rho_B(1-\phi^{local})v_B]}{\partial x}dV_{REV}}_{Cross-diffusion}$$

$$= -\frac{1}{V_{REV}}\int_{V_{REV}}\frac{\partial[\rho_B(1-\phi^{local})]}{\partial t}dV_{REV}, \quad (19a)$$

$$\frac{\partial[\rho_B(1-\phi)]}{\partial t} + \underbrace{\frac{\partial[\rho_B(1-\phi)v_B]}{\partial x}dV_{REV}}_{Self-diffusion} + \underbrace{\frac{1}{V_{REV}}\int_{V_{REV}}\frac{\partial[\rho_A\phi^{local}v_A]}{\partial x}dV_{REV}}_{Cross-diffusion}$$

$$= -\frac{1}{V_{REV}}\int_{V_{REV}}\frac{\partial[\rho_A\phi^{local}]}{\partial t}dV_{REV}, \quad (19b)$$

In a saturated porous medium, a straightforward interpretation of $\rho_A$ and $\rho_B$ may be the density of the fluid phase and that of the solid phase, respectively (Hu et al., 2020). The interpretation of the incorporation of the effects of chemical and thermal processes may not be as straightforward for the observer as they act via Eq. 15 as a linear convolution operation. If we interpret the time-domain convolution operation of THMC waves in the frequency domain, then the chemical and thermal waves can be seen as filters for HM coupling, sharpening or smoothing the waves. The interpretation of THMC waves in terms of a sharpening or smoothing filter analogue is discussed in detail in the companion article (Regenauer-Lieb et al., 2020).

To illustrate the point of choosing a particular time-space scale of observation of THMC waves we first consider the simple homo-entropic flow assumption and choose a classical mechanics point of view. Density is then defined as a function of pressure, temperature, and chemical concentrations by the *Equation of State*. The coupling in Eq. 19 leads to the possible nucleation of Hydro-Mechanical cross-diffusion pressure waves (Hu et al., 2020). Considering that possible thermal processes, $\rho_A$, and $\rho_B$, at the solid-fluid interface may be affected by the local temperature changes, we identify new time-dependent processes at the solid-fluid boundary. The process of heat transport sets two new internal timescales changing the fluid and solid pressure, respectively. Therefore, the thermal process acts as a 'convolution filter' added to the pressure evolution of each phase. Conversely, the pressure diffusion process in the solid and fluid phase triggers two additional timescales in the change of temperature. Now a $3 \times 3$ Thermo-Hydro-Mechanical cross-diffusion formulation can be obtained following the same steps of upscaling from local to a global scale, and the additional 4 timescales correspond to the newly introduced 4 cross-diffusion coefficients. One can arrive at a similar conclusion by considering the interplay between the evolution of pressure distribution and internal chemical reactions.

### 5.0.1 Formulation of the THMC cross-diffusion matrix

The concept of cross-diffusion is well known in chemistry. In a chemical system with just two species $A$ and $B$, for instance, cross-diffusion is the phenomenon, in which a flux of species $A$ is induced by a gradient of species $B$ (Vanag and Epstein,

2009). In more general THMC terms, cross-diffusion is the phenomenon where a gradient of one generalised thermodynamic force drives another generalised thermodynamic flux. Staying with the chemical example of species $A$ and $B$, we have in 1-D:

$$\frac{\partial C_A}{\partial t} = \zeta_A \frac{\partial^2 C_A}{\partial x^2} + L_{AB} \frac{\partial^2 C_B}{\partial x^2} + r_B$$

$$\frac{\partial C_B}{\partial t} = \zeta_B \frac{\partial^2 C_B}{\partial x^2} + L_{BA} \frac{\partial^2 C_A}{\partial x^2} + r_A. \tag{20}$$

where $r_A$ and $r_B$ are the local source terms using the mesoscopic self-diffusion and cross-diffusion decomposition in Eq. (19). Following Eq. (15) we can now generalize (20) to include the full cascade of internal accelerations through multiscale coupling. Cross-diffusion allows coupling of accelerations from one classical REV-scale reaction-diffusion system, defined by the (self-)diffusive length scale $\sqrt{\zeta_{(T,H,M,C)}t}$, to another. The time $t$ is initially defined in this paper from a macroscopic perspective. It is the time for which the macro-scale applied boundary conditions drives a given series of THMC processes. The individual THMC processes may react by a feedback loop between the respective physics of reaction and self-diffusion, often leading to oscillatory behaviour through tight coupling of the system dynamics, thus adding new internal material time-scales. The reaction-diffusion problem is initially fully dynamic but often, after sufficient time has elapsed, it can reach a quasi-static macroscopic response resulting from an oscillatory, or a steady state equilibrium, between the reactive source term and its associated self-diffusion process. In this case the internal material time scale is the time it takes to reach this macro-scale equilibrium (Regenauer-Lieb et al., 2013b, a). The cross-diffusion coefficients, introduced in this paper, enrich the tightly coupled cross-scale self-diffusion-controlled reaction-diffusion processes by linking the gradient of a thermodynamic force $C_j$ of one THMC process to the flux of another kind and thus significantly increases the potential for feedback through additional coupling across scales. The wave operator $\hat{H}_{THMC}$ is now expanded through a fully populated diffusion matrix that includes self-diffusion (diagonal) and cross-diffusion (off-diagonal) coefficients as in:

$$\frac{D\mathbf{C}}{Dt} = \begin{bmatrix} \zeta_T & L_{TH} & L_{TM} & L_{TC} \\ L_{HT} & \zeta_H & L_{HM} & L_{HC} \\ L_{MT} & L_{MH} & \zeta_M & L_{MC} \\ L_{CT} & L_{CH} & L_{CM} & \zeta_C \end{bmatrix} \nabla^2 \mathbf{C} + r_i. \tag{21}$$

The cross-diffusion processes formulate the link between different THMC processes. The cross-diffusion coefficients thereby introduce new cross-scale coupling length- and time scales which are often much smaller than the self-diffusion scales. This is not always the case (Manning, 1970). Hu et al. (2020) show normal examples where cross-diffusion length scales are much smaller than the self-diffusion length scales.

### 5.0.2 Criterion for nucleation of cross-diffusion Waves

A detailed discussion of the criterion for nucleation of cross-diffusion waves and their waveforms can be found in Tsyganov and Biktashev (Tsyganov and Biktashev, 2014). Here, we first summarize the basic method that is well established in the

440 fields of mathematical biology and chemistry and follow on with a discussion of other communities, where the phenomenon of cross-diffusion waves is well-documented under different names.

The criterion for nucleation of cross-diffusion waves relies on assessing the dispersion relation of the eigenvalues of the characteristic matrix of a perturbed cross-diffusion-reaction equation (Vanag and Epstein, 2009). The eigenvalues are functions of the square of the wavenumber of the perturbed state and identify the growth rate of the perturbations. This approach for deriv-

445 ing the mathematical criterion for nucleation of acceleration waves is hence evaluated from a small plane-wave $\epsilon$-perturbation of Eq. (21) with

$$\tilde{\mathbf{C}}(x,t) = \mathbf{C}_0(1+\epsilon)\mathrm{e}^{\lambda t + i(kx)} \tag{22}$$

The characteristic matrix of the thus perturbed Eq. (21) allows assessing the stability of the system. Accordingly, all eigenvalues of the characteristic matrix must be real and positive, and hence the determinant of the matrix must be larger than zero. For

determinants smaller than zero, cross-diffusion waves are expected to propagate as quasi-solitons (Tsyganov and Biktashev, 2014). A worked example for hydromechanical cross-diffusion waves can be found in Hu et al. (2020).

## 6 Soliton versus quasi-soliton solutions

Since cross-diffusion waves in geomaterials are largely unexplored due to the extreme length and time scales encountered in a geosystem, an appreciation of their complex characteristics can be obtained from mathematically similar systems such as

waves in oceans, Lasers, and ice. There is an important difference between solitonic waves and quasi-solitonic cross-diffusion waves. We follow Zakharov et al.'s (Zakharov and Kuznetsov, 1998) definition of solitons and quasi-solitons and identify solutions to the perturbed Eq. 22 of the type:

$$\psi(x,t) = \beta(x - \mathbf{v}t)e^{i\Omega t}, \tag{23}$$

as solitons when the wave amplitude $|\psi(x,t)| = |\beta(x - \mathbf{v}t)|$ propagates without change of form and $\mathbf{v}$ and $\Omega$ are constants.

Quasi-solitons appear as multiscale solutions to Eq. (22) if localized coherent structures defined by true soliton solutions cannot be formed for any type of non-linearity. In the context of the cross-diffusion waves for hydro-poro-mechanical coupling (Hu et al., 2020) real stationary solitons, which propagate with a constant velocity without changing their form, are exact solutions of the Korteweg-deVries equation (Regenauer-Lieb et al., 2013a; Veveakis and Regenauer-Lieb, 2015). They depend on the fluid- and solid self-diffusion coefficients only (Veveakis and Regenauer-Lieb, 2015). Cross-diffusion waves are quasi-solitons

where the two additional cross-diffusion coefficients cannot be eliminated (Hu et al., 2020). These additional time-dependent properties lead to interesting dynamics.

Following dynamic properties of quasi-solitons have been identified (Zakharov et al., 2004): Quasi-solitons live only for a finite time and can be compared to unstable particles in nuclear physics. Unlike true solitons, quasi-solitons loose energy through their oscillatory tails which can have different wavenumbers in the forward and backward direction of their motion. If

the amplitudes of the tails are small, quasi-solitons can be treated as slowly decaying real solitons which lose their energy by radiating quasi-monochromatic waves with wavenumber $k_0$ in the backward direction.

The discrete particle-like behaviour can be explained by their unusual dispersion relation. Quasi-solitons travel with a constant group velocity $\omega'$. When their phase velocity $\omega(k)/k$ exhibits a local minimum at a nonzero wavenumber a gap in the spectrum $\omega(k)$ appears. According to Zakharov et al. (2004), this peculiar discretization of wave energy has been noticed in many disciplines. Different nomenclatures of quasi-solitons have been adopted. In the theoretical physics community, they have been attributed to Cherenkov radiation. In the ice-wave community, they have been called ice-waves with decaying oscillations and in shallow water theory, they have been identified as capillary-gravity waves. If the amplitudes of the quasi-solitons are small and their velocities are close, they obey the non-linear Schrödinger equation and their interaction is elastic. However, the stability and the interaction of large amplitude quasi-solitons are still open questions that cannot be solved analytically. Quasi-solitons move with different velocities and can lead to quasi-solitonic turbulence (Zakharov et al., 2004) when they collide with each other.

In photonics, optical turbulence in the form of sporadic bursts of light (Hammani et al., 2010), have been observed upon the collision of quasi-solitons. In water waves, the same phenomenon is known as 'rogue waves' that seem to appear from nowhere (Akhmediev et al., 2009). Their physics relies on the unusual multiscale energetics of quasi-solitons that can pump energy from the environment to provide a quasi-stationary transport of wave energy from large to small scales. Earlier, we have introduced a simple 'convolution filter' interpretation of THMC coupling. In this sense, the interaction of THMC-waves may be seen as an extreme form of sharpening filter that can generate rogue waves.

The independent choice of a reference system such as discussed in the convolution filter analogy also applies to the energy carried by the waves. If we choose for instance an observer of hydro-mechanical waves, the inverse energy cascade from THMC wave action from small to large-scale and the direct energy cascade from large to small-scales (Zakharov et al., 2004) allows the identification of rogue waves, if they occur. The phenomenon of the collision, merging, and the collapse of quasi-solitons may provide a mechanism for a bi-directional THMC energy cascade that leads to earthquakes as a form of solid-state turbulence as discussed in the application in the companion article (Regenauer-Lieb et al., 2020).

## 7   Conclusions

This paper has introduced three important innovations for modelling THMC instabilities: (i) a multiscale extension of the theory of thermodynamics of irreversible processes to include dynamic events by using a meso-macro-scale model; (ii) a generalization of the theory of cross-diffusion waves from chemical systems to generalised THMC thermodynamic-force flux pairs; (iii) a transfer of knowledge from classical quantum mechanics to characterise any system at a larger scale in order to deal with the discreteness of multiscale material behaviour.

(i) We have shown that cross-diffusion waves in THMC systems can be decomposed into cross-diffusional $S$- and $P$- acceleration waves and have discussed a THMC multiphysics implementation, where cross-diffusion waves appear as quasi-soliton waves for critical conditions identified from a perturbed Eq. (21). These waves radiate energy away from meso-scale sources that are incompatible with the overall macro-scale stress gradients with a complex, reflection, and interaction behaviour into the far field. This finding overcomes the problem of unbounded solutions encountered in the classical solid mechanical theory

of localisation (Benallal and Bigoni, 2004) where acceleration waves are modelled on the basis of coupled thermomechanics reaction-diffusion equations without the cross-diffusion term. The necessity of decomposing the macro-scale reactive source term into a meso-scale reaction-diffusion couple was discovered recently by using mixture theory (Hu et al., 2020).

(ii) The multiscale approach can be encapsulated in a concise fully populated self/ cross-diffusion matrix (Eq. 21). The theory is based on an extension of the chemical systems to generalised linear thermo-dynamic force-flux pairs at the meso-
510 length/time scale. These are found to nucleate cross-diffusion waves under critical conditions and replicate nonlinear behaviour for the long time/spatial macro-scale. This occurs when cross-diffusion waves converge to standing-wave solutions when their radiative tails become vanishingly small such as in the non-linear Schrödinger equation or the Korteweg-deVries equation. The approach reveals that instabilities based on the shear and volumetric response of the material at the meso-scale are fundamentally important and have been overlooked. We have shown that incompatibilities of meso-scale accelerations with the overall
stress field lead to the nucleation of cross-diffusion waves which travel in an unstable particle-like state with characteristic material velocities $c$ defined by the competition of meso-scale reaction-diffusion processes at the propagating wavefront. The physics of this phenomenon is discussed further in Regenauer-Lieb et al. (2020). These velocities characterise the progress of internal material timescales for the formation of multiscale space/time dissipative structures and are characteristic properties for the dynamic behaviour of a given material. These internal material clocks are here introduced as a multiscale THMC
cascade for coupling the physics of the very small to the very large.

(iii) Cross-diffusion waves have first been discovered for interactions in quantum mechanics such as in photonics where they show anomalous dispersion patterns that, unlike solitons radiate energy in the form of oscillatory (Cherenkov)-tails (Zakharov and Kuznetsov, 1998; Paschotta, 2008). This unusual energy radiation property differentiates them from solitons as quasi-solitons. Since they assume an unstable particle-like state (see the example in Fig. 2), the reflections and collisions, when
they happen, can lead to a variety of responses (Biktashev and Tsyganov, 2016; Zakharov et al., 2004; Lioubashevski et al., 1996). Despite their nucleation through discrete internal micro-dissipative mechanisms, cross-diffusional waves also show proper soliton wave-like behaviour and can penetrate through each other and reflect from boundaries. However, unlike true solitons, their amplitude and speed are not controlled by initial conditions but by material properties (Tsyganov and Biktashev, 2014). The effect of cross-diffusion is to trigger cross-diffusion waves for critical conditions. They form by THMC-feedback
as discrete material instabilities which can be either observed as a local, discrete failure or as damage waves.

They are found to propagate and reflect from boundaries in a multiscale energy cascade which in extreme cases can lead to the formation of a local turbulence phenomenon that seems to appear from nowhere. The phenomenon of wave sampling of energy into ultra-localized events is well known in many disciplines and appears in oceans as 'rogue waves' (Akhmediev et al., 2009) or in laser physics as sudden 'light-bursts' (Solli et al., 2007) where the interesting multiscale spectral content of quasi-
535 solitons, their unstable collisional nature and their capability to focus energy from the large-scale to feed local scale 'rogue waves' (Zakharov et al., 2004; Akhmediev et al., 2009) is already well documented. In the companion article (Regenauer-Lieb et al., 2020) we show that the cross-diffusion formulation also follows from a small meso-scale perspective, when looking at the large macro-scale. A particular application will be the discussion on a potential application to earthquake source mechanisms both as a precursor and as well as the main- and post-seismic creep event.

## 8  Acknowledgments

This work was supported by the Australian Research Council (ARC DP170104550, DP170104557, LP170100233) and the strategic SPF01 fund of UNSW, Sydney. We would also like to acknowledge valuable constructive feedback of Angelo De Santis and two anonymous reviewers. We thank Emily and Anders Crofoot of Castepoint Station (NZ) for their hospitality and access to the beautiful rocks on their property. FInally, we would like to acknowledge support from the Central Analytical Research Facility (CARF) of QUT.

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
