# Peer review of "Cross-Diffusion Waves resulting from Multiscale, Multi-Physics Instabilities: Theory"

_Solid Earth, 2020_

## Referee Comment (RC1) · Angelo De Santis (Referee) · 28 Sep 2020

This manuscript proposes an approach for coupling processes that the authors define "Multiphysics" because due to different kinds of instabilities. Although the work looks intriguing in approaching the problem with a complete and integrated view, in my opinion, there are some parts that are not yet clear.

The main problem I found in reviewing this paper is that the authors sometime, instead of explaining directly the question, prefer to resort to another companion paper that, shamefacedly, is not yet available (at least to me). In general, I would suggest to describe the most important passages of that companion paper (a dedicated section

would be useful), so also the present work can be self-consistent and explanatory.

Another important aspect is that if the new theory wants to replace the conventional present one, the latter should be concisely described and the critical issues pointed out, so justifying the replacement with the new one. This aspect is neglected.

For the rest, the paper deserves attention and, once all critical points are solved, could be of great interes by the scientific community.

Other points/doubts/comments

Lines 10-11. The final sentence of the Abstract is not fully clear. The same problem is found in the main text.

Lines 12 and following. The Introduction is too short and poor.

Lines 50-52. Very interesting, ma not clear at all from this work. This sentence remains suspended, Also because of the reference to the companion paper which is not yet available.

Lines 63-64. This should be better explained and extended. Otherwise, also this sentence remains suspended.

Line 91. Eq. 1. For dimensional coherence, the body force f should be alone, not multiplied by ro. Consequently, adapt in the following part of the text.

Line 104. definition of I as Identity matrix should be moved below (after all this series of expressions).

Line 119. Putting in the same name "elastic" and "plastic" looks contradictory. What do you mean? that the strain rate can be in different conditions? But this would be said in the text.

Line 287, Eq. 21a. Is here missing a dV_REV?

Lines 378-380. I cannot have access to this other work (Regenauer-Lieb et al., 2020).

It would have interesting to have a look at it. Otherwise, I suggest to mention the most significant passages of that work.

Lines 403-404. Not convincing and neither clear. This point should be more clarified in this work (and not only referred to the companion paper). For instance, how is the new theory better than the conventional one that describes earthquake processes?

---

## Referee Comment (RC2) · Jean Paul Ampuero (Referee) · 30 Sep 2020

This paper presents expectations about patterns in the solutions of a set of equations that are proposed to be relevant to Earth processes. The authors introduce equations of THMC coupled processes that include cross-diffusion terms (gradients of one quantity induce diffusion of another quantity) and then leverage on existing literature on the topic to anticipate the rich patterns the solutions could develop. No new simulations nor values of relevant non-dimensional parameters in the context of Earth processes are presented to support the proposed ideas. Hence no quantitative predictions are made, based on the model proposed, that could serve as testable hypotheses for future field

or laboratory observations.

This was a very difficult paper to review because of the poor quality of the writing. While grammar and spelling are perfect, the logical structure of the paper is deficient, at times hard to discern. Here are concrete examples:

1. The introduction opens with an argumentum ad verecundiam (akin to "a Nobel prize winner said . . ."). One would rather expect arguments to be supported by peer-reviewed references.

2. The introduction is 4 sentences long, cites no references, is obscure and poorly interconnected, and overall fails to accomplish the basic goals of an introduction section. What is the Earth science problem addressed? What observations and open questions motivate this work? Why does it matter for an Earth scientist? What is the state-of-the-art in understanding these phenomena? What is the research gap to be filled? One example: the topic of reactive transport is abundantly treated in the Earth science literature, that thread could be explored and cited here.

3. The concept of cross-diffusion is so important that it is in the title, but it is not defined until section 5.

4. Moving goalposts: Near line 50, the reader gets the impression that the paper will be about earthquake precursors. An exciting prospect, indeed. But there is no substantial treatment of earthquake processes in the rest of the paper. The earthquake topic appears, almost as an afterthought, in two sentences near the end of the paper. Most of the paper seems to be about compaction bands.

5. The choice of words (like "postulate" instead of "hypothesize" in line 53) gives the sense of the work being a theoretical musing rather than an effort to develop testable hypotheses. Observational and experimental challenges are described later in the paper, as well as references to ongoing related work (in review?), but placing those instead in the introduction section would help the reader understand the aim and scope

of the paper.

6. Line 62: If you are really after earthquake precursors that are analogous to travelling waves, there may be interesting connections to draw between your models and slow slip phenomena (abundant literature on that topic, including observations, experiments and modeling).

7. Line 80, "We call this extension (geo-)wave mechanics due to its mathematical similarity with quantum physics approaches": unfortunately, such analogy to quantum mechanics is not elaborated anywhere in the paper.

8. Unclear notations. For instance, in equation 15, why a subscript i in the Laplacian and in the coordinates (x,y,z), instead of applying the same Laplacian to N physical quantities that have a subscript i ?

9. Section 5 is basically section 4 of Hu et al (2020). Without loss of clarity or emphasis, this section can be substantially shortened by referring to those previously published results.

---

## Author Comment (AC1) · 23 Oct 2020

We would like to thank the reviewers for the tremendous effort and careful review of this paper. We realize that the paper touches a number of challenging fundamental issues and is therefore more difficult and time-consuming to review than the average publication. We are extremely grateful for the patience and detailed feedback providing useful criticism and suggestion for modifications.

The reviews of the original paper suffered from two significant issues.

By not having the second paper available both reviewers were missing the important

application of this paper presented in part 2. Unfortunately, the second part was not yet available to the reviewers at the time of their reviews as the submission had been delayed and the paper had not yet been released for viewing. The second part is now accessible to the reviewers and available on https://doi.org/10.5194/se-2020-149, in review, 2020.

Another significant problem was the deficient introduction which did not specify why this paper was focusing just on the theory aspect and how the two papers join together. The original intention was indeed to submit both papers in one cast but delays in the writing of the second manuscript led to a placeholder introduction in part 1. We apologize for this shortcoming.

The introduction has now been entirely rewritten. It includes the necessary link to the second part. It also lists the relevant background in terms of the GEOPROC themes, references, and prior work in the development of the new theory and explains the innovations in this paper that overcome the issues encountered in earlier attempts to address the dynamic regime of acceleration waves. The introduction also provides a short overview of the sections to come and in conjunction with the update and rewritten conclusion should present a clear logical structure of the paper.

This paper was written with the specific aim to derive and explain the new theory in detail which did not receive any criticism. Please find below the address of the specific points raised by the reviewers.

Angelo De Santis (Referee) general comment: This manuscript proposes an approach for coupling processes that the authors define "Multiphysics" because due to different kinds of instabilities. Although the work looks intriguing in approaching the problem with a complete and integrated view, in my opin- ion, there are some parts that are not yet clear. The main problem I found in reviewing this paper is that the authors sometime, instead of explaining directly the question, prefer to resort to another companion paper that, shamefacedly, is not yet available (at least to me). In general, I would suggest to

[Figure]

describe the most important passages of that companion paper (a dedicated section would be useful), so also the present work can be self-consistent and explanatory.

Reply: That is an excellent suggestion. We have modified the introduction followed by a section summarizing the specific introduction to the second paper.

Angelo De Santis (Referee) general comment: Another important aspect is that if the new theory wants to replace the conventional present one, the latter should be concisely described and the critical issues pointed out, so justifying the replacement with the new one. This aspect is neglected.

Reply: We do not challenge the existing rate and state variable friction theory for earthquakes but have now explained much better in the introduction why it is necessary to introduce a new theory to overcome the shortfalls of the conventional theoretical and applied mechanics approach that has not yet been applied to earthquakes. Specifically, we contrast our approach in the introduction to the existing approach. The relevant section in the introduction reads:

" Before discussing a possible application of the new theory to the processes of earthquakes and faulting in our companion paper \citep{Geoproc2020b}, here we present a transdisciplinary approach bridging the gap between observations of instabilities from the molecular scale to the very large scale. The theory in this paper is written using approaches familiar to the theoretical and applied mechanics community. The original work is based on the 1960's work \citep{Hill_acceleration} building the foundation of theoretical approaches to localisation criteria, via the so-called acoustic tensor criterion, widely used in the engineering community \citep{Rudnicki75}. The approach focuses on standing wave quasi-static solutions based on vanishing speeds of acceleration waves. Surprisingly, little effort has been made to explore the rich wave field of the corresponding traveling wave solutions, probably because dynamic events are only of academic interest to the engineering plasticity community that focusses mainly on developing safety standards as well as limit analysis and design. A notable exception is

the work of \cite{Benallal2004} who found that under dynamic conditions, unbounded growth of perturbations can be found in the short wavelength regime with divergence growth. This calls for an extension to the theoretical work of \cite{Hill_acceleration} which is presented here.

The dynamic field is, however, of special interest to the researcher in the area of earthquake and faulting instabilities. The state of the art in this field is defined by the influential experimental work of \cite{Dieterich79} including the early work on the application of the rate and state variable friction approach to earthquakes \citep{Tse1986}. The approach based on these laboratory-derived constitutive equations has reached a mature stage, and no attempt is made here to compare the rich field of findings with the present theory. We approach the problem from an entirely different angle through theoretical investigation of the mathematical solutions of the system of coupled partial differential THMC equations that deliver wave solutions with short-wavelength instabilities. In the course of developing the new approach, we describe wave physics phenomena that have previously not been reported in geophysical solids but are well known in a range of different fields from quantum systems to ocean waves \citep{Zakharov2004}. It is fair to say that the theory is rather in its infancy state, and special care needs to be taken before considering a direct application to the aforementioned systems. The first part, therefore, presents the theoretical derivation, and the second part delves into possible applications and proposed experimental tests to verify the applicability of the theory.."

Angelo De Santis (Referee) Specific comments Other points/doubts/comments Lines 10-11. The final sentence of the Abstract is not fully clear. The same problem is found in the main text.

Reply: This sentence is a preview on part 2 we now refer to part 2 and also include it in the abstract by saying

"Part 2 proposes an application to earthquakes showing that for extreme conditions,

cross-diffusion waves can lead to an energy cascade connecting large and small-scales and cause solid-state turbulence.":

Angelo De Santis (Referee) Specific comments Lines 12 and following. The Introduction is too short and poor.

Reply: corrected

Angelo De Santis (Referee) Specific comments Lines 50-52. Very interesting, ma not clear at all from this work. This sentence remains suspended, Also because of the reference to the companion paper which is not yet available.

Reply: We now refer to the second paper explicitly "We will discuss such possible precursor phenomena for earthquakes in part 2 \citep{Geoproc2020b}."

Angelo De Santis (Referee) Specific comments Lines 63-64. This should be better explained and extended. Otherwise, also this sentence remains suspended.

Reply: We now refer to the second paper explicitly "In the supplementary material of part 2 \citep{Geoproc2020b} we will discuss possible experimental tests of the precursor phenomena."

Angelo De Santis (Referee) Specific comments Line 91. Eq. 1. For dimensional coherence, the body force f should be alone, not multiplied by ro. Consequently, adapt in the following part of the text. Reply: yes, the body force needs to have rho removed, this is now adapted in the text Line 104. definition of I as Identity matrix should be moved below (after all this series of expressions).

Reply: the explanation is moved to the appropriate place

Angelo De Santis (Referee) Specific comments Line 119. Putting in the same name "elastic" and "plastic" looks contradictory. What do you mean? that the strain rate can be in different conditions? But this would be said in the text.

Reply: the incorporation of the yield phenomenon through the elasto-viscoplastic

compliance is now better explained. We have added following text: " in this case $\boldsymbol{\dot \epsilon}'$ denotes the deviatoric strain rate which in the purely elastic case before yield is $\dot{\boldsymbol{\epsilon}}'=\dot{\boldsymbol{\epsilon}}'_{{e}}$ becoming post-yield the elasto-viscoplastic strain-rate defined by $\dot{\boldsymbol{\epsilon}}'=\dot{\boldsymbol{\epsilon}}'_{{e}}+\dot{\boldsymbol{\epsilon}}'_{{vp}}$. ${\dot \epsilon_0}$ is the equivalent elasto-viscoplastic volumetric strain-rate."

Line 287, Eq. 21a. Is here missing a dV_REV? Reply: yes it is missing and has been replaced

Angelo De Santis (Referee) Specific comments Lines 378-380. I cannot have access to this other work (Regenauer-Lieb et al., 2020) It would have interesting to have a look at it. Otherwise, I suggest mentioning the most significant passages of that work.

Reply: The manuscript is now available on https://doi.org/10.5194/se-2020-149 , in review, 2020.

Angelo De Santis (Referee) Specific comments Lines 403-404. Not convincing and neither clear. This point should be more clarified in this work (and not only referred to the companion paper). For instance, how is the new theory better than the conventional one that describes earthquake processes?

Reply: We have rewritten the entire conclusion in order to link up to the new introduction and explain that the meso-scale processes have been overlooked in conventional earthquakes and solid mechanics theories.
* * *

---

## Author Comment (AC2) · 23 Oct 2020

We would like to thank the reviewers for the tremendous effort and careful review of this paper. We realize that the paper touches a number of challenging fundamental issues and is therefore more difficult and time-consuming to review than the average publication. We are extremely grateful for the patience and detailed feedback providing useful criticism and suggestion for modifications.

The reviews of the original paper suffered from two significant issues.

By not having the second paper available both reviewers were missing the important

application of this paper presented in part 2. Unfortunately, the second part was not yet available to the reviewers at the time of their reviews as the submission had been delayed and the paper had not yet been released for viewing. The second part is now accessible to the reviewers and available on https://doi.org/10.5194/se-2020-149, in review, 2020.

Another significant problem was the deficient introduction which did not specify why this paper was focusing just on the theory aspect and how the two papers join together. The original intention was indeed to submit both papers in one cast but delays in the writing of the second manuscript led to a placeholder introduction in part 1. We apologize for this shortcoming.

The introduction has now been entirely rewritten. It includes the necessary link to the second part. It also lists the relevant background in terms of the GEOPROC themes, references and prior work in the development of the new theory and explains the innovations in this paper that overcome the issues encountered in earlier attempts to address the dynamic regime of acceleration waves. The introduction also provides a short overview of the sections to come and in conjunction with the update and rewritten conclusion should present a clear logical structure of the paper.

This paper was written with the specific aim to derive and explain the new theory in detail which did not receive any criticism. Please find below the address of the specific points raised by the reviewers.

Jean-Paul Ampuero (Referee) General Comment: This paper presents expectations about patterns in the solutions of a set of equations that are proposed to be relevant to Earth processes. The authors introduce equations of THMC coupled processes that include cross-diffusion terms (gradients of one quantity induce diffusion of another quantity) and then leverage on existing literature on the topic to anticipate the rich patterns the solutions could develop. No new simulations nor values of relevant non-dimensional parameters in the context of Earth processes are presented to sup-

[Figure]

port the proposed ideas. Hence no quantitative predictions are made, based on the model proposed, that could serve as testable hypotheses for future field or laboratory observations.

Reply: We acknowledge that the introduction jumped straight into the topic without giving the reader a hands-on introduction to the motivation, the state of the art in the various fields and the important innovation in this paper. We have therefore completely rewritten the introduction to give a much clearer picture of why we did not in this paper specify non-dimensional parameters relevant to Earth processes.

The analyses expected by the reviewer have indeed been our first approach which should have been cited in order to explain why a numerical non-dimensional analysis of non-dimensional exploration is not fully satisfactory. We now explain in the introduction the background of these papers. We were able to use the numerical solution of the system of coupled pde's using non-dimensional parameters to show that NVTS events (Poulet, Veveakis et al. 2014) and geological faults (Poulet, Veveakis et al. 2014) can be modelled by the approach. However, a sound theoretical description and interpretation of the local processes resulting in the interesting macroscopic phenomena has been lacking. Numerical solutions of coupled THMC-equation have a notorious problem of modelling instabilities out of a perturbed state and while sophisticated solutions (arc continuation method) have been developed a proper understanding of the instabilities and analytical criterion for the onset of dynamic instabilities did not exist prior to this work. A very important problem is the non-uniqueness of the solutions and the associated uncertainty relationships which requires new solution strategies.

All three shortcomings are overcome in the present contribution. We acknowledge, however, that the new theory is in its infancy state and special care needs to be taken before applying it to real-world systems. The first part is therefore presenting the theoretical derivation and the second part delves into possible applications and proposed experimental tests to verify the applicability of the theory to Earth processes

In this paper we focus on following innovations that cannot be found elsewhere in the literature: (i) a multiscale extension of the theory of thermodynamics of irreversible processes to include dynamic events by using a meso-scale - macro-scale model discussed in part 1 and illustrated in further details in part 2; (ii) a generalization of the theory of cross-diffusion waves from chemical systems to generalized THMC thermodynamic-force flux pairs; (iii) a transfer of knowledge from classical quantum mechanics to characterise any system at a larger scale in order to deal with the discreteness of multiscale material behaviour and the quantum-like uncertainty relationships. All of these aspects are listed in the introduction and. the conclusion summarizes the theoretical findings of this paper

References to our earlier work on non-dimensional analyses of NVTS events and geological faulting in poromechanic, reactive systems now added to the introduction Poulet, T., E. Veveakis, K. Regenauer-Lieb and D. A. Yuen (2014). "Thermo-Poro-Mechanics of chemically active creeping faults. 3: The role of Serpentinite in Episodic Tremor and Slip sequences, and transition to chaos." Journal of Geophysical Research: Solid Earth 119(6): 4606–4625 Poulet, T., M. Veveakis, M. Herwegh, T. Buckingham and K. Regenauer-Lieb (2014). "Modeling episodic fluid-release events in the ductile carbonates of the Glarus thrust." Geophysical Research Letters 41(20): 7121-7128.

Jean-Paul Ampuero (Referee) detailed comments: This was a very difficult paper to review because of the poor quality of the writing. While grammar and spelling are perfect, the logical structure of the paper is deficient, at times hard to discern. Here are concrete examples: 1. The introduction opens with an argumentum ad verecundiam (akin to "a Nobel prize winner said . . ."). One would rather expect arguments to be supported by peer- reviewed references. 2. The introduction is 4 sentences long, cites no references, is obscure and poorly interconnected, and overall fails to accomplish the basic goals of an introduction sec- tion. What is the Earth science problem addressed? What observations and open questions motivate this work? Why does it matter for an Earth scientist? What is the state-of-the-art in understanding these phenomena?

What is the research gap to be filled? One example: the topic of reactive transport is abundantly treated in the Earth science literature, that thread could be explored and cited here. 3. The concept of cross-diffusion is so important that it is in the title, but it is not defined until section 5.

Reply: We fully acknowledge the deficiency of the original introduction. The new introduction and the con-jointly rewritten conclusion should solve all of the above issues also putting this paper into the rich context of the GEOPROC conference series.

Jean-Paul Ampuero (Referee) detailed comments: 4. Moving goalposts: Near line 50, the reader gets the impression that the paper will be about earthquake precursors. An exciting prospect, indeed. But there is no substantial treatment of earthquake processes in the rest of the paper. The earthquake topic appears, almost as an afterthought, in two sentences near the end of the paper. Most of the paper seems to be about compaction bands.

Reply: This misconception is due to the poor quality of the original introduction and the unavailability of the second paper. The first reviewer also stumbled over this. The rationale in the introduction and the second paper should alleviate this concern. The goalposts of the first and second paper are now clearly articulated in the introduction. " It is fair to say that the theory is rather in its infancy state, and special care needs to be taken before considering a direct application to the aforementioned systems. The first part therefore presents the theoretical derivation, and the second part delves into possible applications and proposed experimental tests to verify the applicability of the theory.".

Jean-Paul Ampuero (Referee) detailed comments: 5. The choice of words (like "postulate" instead of "hypothesize" in line 53) gives the sense of the work being a theoretical musing rather than an effort to develop testable hypotheses. Observational and experimental challenges are described later in the paper, as well as references to ongoing related work (in review?), but placing those instead in the introduction section would

help the reader understand the aim and scope of the paper.

Reply: We exchanged the word postulate by hypothesis

Jean-Paul Ampuero (Referee) detailed comments: 6. Line 62: If you are really after earthquake precursors that are analogous to travelling waves, there may be interesting connections to draw between your models and slow slip phenomena (abundant literature on that topic, including observations, experiments and modeling).

Reply: very good suggestion! We did not present our earlier work where we are applying coupled Multiphysics pde's numerically (Poulet, Veveakis et al. 2014) to show that they can replicate slow slip events. The earlier work is critically reviewed in part 2 but following suggestions we also highlight the discussion of the second paper in the revised version in the introduction of part 1.

Jean-Paul Ampuero (Referee) detailed comments: 7. Line 80, "We call this extension (geo-)wave mechanics due to its mathematical similarity with quantum physics approaches": unfortunately, such analogy to quantum mechanics is not elaborated anywhere in the paper.

Reply: Right! This needs further explanations. We have removed the notation of geo-wave mechanics as it distracts from the main points in this paper. The paper is already rich in theoretical concepts and following this strand may be interesting but possibly leading too far astray. It may be more interesting to the theoretical physics community. We felt that we should, however, acknowledge somewhere inspiration from the quantum approach.

The new introduction, therefore, promises a discussion of the link to quantum statistical approaches that allow overcoming the problem of dealing with discrete patterns and the uncertainty relationship due to path dependence of the integration of the thermodynamic system (incomplete heat and work differentials). The revised conclusion summarizes this point succinctly. Part 2 provides around equation 29a+b a full discussion of the uncertainty principles of quantum systems applied to the systems at a larger scale which stems from the analogy of Schrödinger's equation (equation 28). The formal analogy to Schrödinger's equation is not mentioned, however, for the reasons stated above. While this analogy is an interesting discussion it does not aid the understanding of the paper.

Jean-Paul Ampuero (Referee) detailed comments: 8. Unclear notations. For instance, in equation 15, why a subscript i in the Laplacian and in the coordinates (x,y,z), instead of applying the same Laplacian to N physical quantities that have a subscript i ?

Reply: Correct the subscript is unclear it only applies to the N physical diffusion processes. This has been corrected in the revision.

Jean-Paul Ampuero (Referee) detailed comments: 9. Section 5 is basically section 4 of Hu et al (2020). Without loss of clarity or emphasis, this section can be substantially shortened by referring to those previously published results.

Reply: We have shortened the mixture theory section in the revision. " In the following discussion, we generalise the discussion of the meso-scale mass exchange processes using mixture theory applied to HM coupling as presented in \cite{Hu2019b}. We show the physics of cross-diffusion follows from a reactive source term at the macroscale that requests a cross-diffusion term at the meso-scale for thermodynamic consistency. The full derivation is found in \cite{Hu2019b}. Here we summarize the main conclusion from the mixture theory analysis."

---

## Referee Report (RR1)

[referee-annotated manuscript omitted]

---

## Author Response (AR2)

*Reply to the reviewer's comments:*

*We would like to thank Angelo De Santis and the anonymous reviewer for the additional careful review of this paper. We appreciate that the review by Angelo De Santis allows us to make the new approach available to a much broader audience and we are again, extremely grateful for the patience and detailed feedback providing useful criticism and suggestion for modifications.*

*We are also very grateful for the anonymous second review in particular for highlighting potential misunderstanding stemming from the original title of the article and the need for explicitly mentioning the difference between Hu et al 2020 and the generalised multiscale approach. We feel that the additional explanations are important and were indeed lacking in the earlier version.*

*Please find below the address of the specific points raised by the reviewers.*

*Yours kindly Klaus Regenauer-Lieb*

**Annotated NOTES for* "Cross-Diffusion Waves as a trigger for Multiscale, Multi-Physics Instabilities: Theory" *by* Klaus Regenauer-Lieb et al.**

**Angelo De Santis (Referee)**

**Line 1:** We propose a non-local, meso-scale approach for coupling multiphysics processes across the scales. I find this sentence a little contradictory: "non-local meso-scale approach"... that deals with "...processes across the scales." I think a more clear statement is needed, especially  because it is the first!

*Reply: Thanks for pointing this potential confusion out: The term "non-local" only speaks to a special applied mathematics or  mechanics and physics of solids community and "meso-scale" has different meanings in different communities and is thus confusing.  The sentence is now simplified to "We propose a multi-scale approach for coupling* multiphysics *processes across the scales."*

The term nonlocal reaction-diffusion equations is also now defined much later in the text (line 353 ff in the revision with highlights of changes) and a reference has been added.

**Line 54:** strange that the "early work" is referred by a more recent paper (Tse and Rice 1986) than Dieterich

*Reply: The qualifier "early" is removed*

**Line 59:** What do you mean for "geophysical solids"? I could understand it, but as a geophysicist it is unusual. Do you simply mean "earth science" or "solid earth" or what? I do not say that this term is wrong but it is not commonly used in geophysycs.

*Reply: Changed to solid earth community*

**Line 65 following:** How far is your theory from the accepted continuum mechanics? Which are the differences and similarities? At first glance, the main differences are: a) one concerns with continuous processes, while the other concerns with discontinuous, intermittent, puntual processes; and b) you add also thermal and chemical  aspects. But nothing explicit and clear is said. I think one-two sentences, at least in the Conclusions, would be of great help for any potential reader.

*Reply: The classical theory of standing waves accepted in continuum mechanics is a special case of the present approach which comes out for a relatively narrow parameter space. This indicates perhaps that processes in nature have a much richer solution space than anticipated by the classical theory of localisation. This shortcoming of the classical theory (overlooking two different forms of instabilities) is not addressed in the current paper as it requires additional derivations.*

*The current manuscript only addresses the two main problem in the classical approach. The first is that the classical theory disregards propagating waves in the solution space, the second is that the solution is mathematically ill-posed due to the infinite response of strain-rate in shear, or overpressure in compression, and the third that is axiomatic and not based on physics processes as is preferred in geosciences because of the need for extrapolation on longer time scales inaccessible to the observer.*

*The standard approach in the solid mechanics community is to either add an empirical material length scale (e.g. related to strain gradients) or use numerical diffusion to capture the singularities. Only few researchers have attempted to regularise the approach by referring to physics phenomena through introduction of a self-diffusion process or a chemical reaction term or both. This is a hot topic and now better explained through an additional paragraph on line 56 in the marked up revision.* We also come back to the point on line 353 ff.

**Line 71:** here you say that you investigate the meso-scale ... from the... solution of the macro-scale. In short, you affirm that the study of a scale has effect in understanding the features of another scale. This is not obvious and should be better explained and clarified

*Reply: This has now been amended and additional text added*

**Line 71:** his sentence is not clear:
"the scale .... is defined by ...[their] own ... scales". Would do you mean:
"we propose that each of the THMC processes is defined by [its] own characteristic diffusion time/length scales"?
In addition, I have clear idea about diffusion time (in terms of the solution of diffusion equation), but how do you define the length scales? This also is not clear.

*Reply: Changed to "In order to define the separation between the meso- and macro-scale of a THMC coupled problem, we propose that the scale for each of the THMC-processes is defined by its own characteristic diffusion time/length scales \citep{JCSMD1}. The THMC diffusion length scale is thereby related to the time scale of a considered THMC process as defined by the proportionality to the square root of the diffusivity multiplied by the process time."*

**Figure 3 caption:** In the main text you speak about extended length scales: here the length scale is rather short. How do you combine the two points?
In addition: I notice with respect to a majority of lines there are also some hortogonal (and other inclined) lines, which are not reminiscent of a standing wave or are difficult to be explained.

*Reply: This is well spotted. Results of our detailed field work are now available and cited in the caption. The standing waves are the exception rather than the rule both from the linear stability analysis performed by us (to be discussed in a forthcoming contribution) and as well from geological observations, where irregularly spaced deformation bands are more frequently encountered. The standing waves are normally found far away from the faults and form the smallest spacing (around 10 cm) in our field example. The figure has been replaced and more explanation is added to the caption including reference to the new publication and the question of scale is now clarified in the caption.*

**Line 326:** it is not clear. Do you mean "... H_THMC corresponding to the conditions of Table 1"? Or what?

*Reply: In order to generalize the approach, we propose that all reaction-diffusion equations in Table 1 are strongly coupled. We construct a composite multi-scale THMC diffusion wave operator ^HTHMC from the four reaction-diffusion equations in Table 1.*

$$\hat{H}_{THMC} = -\zeta_i \sum_{i=1}^{N} \nabla^2,$$

where $\nabla^2 = \frac{\partial^2}{\partial x^2} + \frac{\partial^2}{\partial y^2} + \frac{\partial^2}{\partial z^2}$ and $i$ refers to the individual thermodynamic THMC process.

*The wave operator is now also discussed in further details on line 353 following.*

**Line 393:** here t is generic. How do you define numerically the length scales? Do you replace it by the diffusione time? Or what?

*Reply: The time t is now defined on line 430 ff in the revised annotated manuscript*

**Line 399:** I do not understand: you are comparing "cross-diffusion length/time scales" with "self-diffusion length scales". The former includes space and time scales, the latter only space scales.

*Reply: It is now consistent*

**Line 501:**This sentence comes out of the blue. You should refer to the second part, where, I presume ( I did not yet read it), you provide more convincing arguments. I would also suggest some more clarifying sentence concerning this point.

*Reply: The sentence is now clarified:* see line 553 in the annotated revisions.

In addition all minor typographical errors have been corrected.

**Response to Anonymous(Referee) comments**

**Suggestions for revision or reasons for rejection (will be published if the paper is accepted for final publication)**
This constitutes my review of the Regenauer-Lieb et al manuscript "Cross Diffusion Waves …: Theory. This is paper 1 of a two-paper series. I enjoyed reading this paper and feel that it should be accepted with minor revisions. I detail some minor corrections below (these are minor). The paper is a good companion to the Solid Earth Special Issue deriving from the recent GEOPROC conference in Ultrecht.

Two aspects present themselves which the authors should concern themselves in the revisions. First, much of the theory and discussion in this paper mirrors the treatment in the recent Hu et al. paper (J. Mechanics Phys Solids 135, 2020). The authors should discuss how the present paper differs in content from the Hu et al. paper.

*Reply: The revised version now explains the present work in relation to the Hu et al.(2020) paper. Differences in the present work are closing the link (and spelling out the differences) to the classical established theory of Hill et al. and following authors (new paragraph following line 56 in the marked-up revised manuscript as well as the paragraph following line 353), establishing the case for the generalized multiscale THMC problem and linking up with other cross-diffusion formulations from different disciplines. The intellectual contribution of the derivation of the cross-diffusion term from two-scale mixture theory in Hu et al.(2020) is now also more explicit. A paragraph has been added to clarify the rationale to repeat the mixture theory approach in generalised form here (line 365 in marked up revision). This approach constitutes an important difference to other attempts (including our own) at closing the ill-posed reaction-diffusion problem through ad-hoc assumptions.*

Second, I feel that the "cross-diffusion" aspect of the wave propagation is confusing and potentially ill-suited to describe the onset of THMC instabilities. The authors accurately describe the "cross diffusion" matrix in analogy with the cross diffusion (non diagonal) terms arising in chemical diffusion as occurs in concentrated brines or melts e.g. These represent "linear" couplings between fluxes and forces in the Onsager sense. As such the terms in the matrix do not describe the (by necessity, as described by Prigogine) non-linear couplings that arise from 1. Non equilibrium conditions and 2. The existence of coupling feedbacks. I think its important to distinguish excitability (nonlinear growth of perturbations) which would be described by the reaction rate terms in their Table 1 or in their equations (21). I think this is well discussed in the Tsyganiov reference the authors include. Thus I question the linkage of "cross diffusion" with instability given in the title.
The same would be true of the "acoustic tensor " plasticity approach of Hill and Rudnicki and Rice – they discuss the "uniqueness" of plastic instabilities but a priori assume the "existence".

The phenomena discussed in this paper are related to the compaction/density waves discussed by Dan McKenzie in 1984 and in that case, the instabilities require a nonlinear coupling of porosity with permeability – surely the linear requirement of the cross diffusional matrix (which is explained better in the HU et al paper) preclude this type of instability? The authors may wish to elucidate these concerns, and hopefully these don't muddy the waters but lead to a clarification. Certainly the non diagonal terms in the THMC matrix in the authors Table 1 are important considerations in the propagation of soliton-like waves, but by themselves they don't describe the onset of instability leading to the development of such waves???

Or am I missing something?

*Reply: We would like to thank the reviewer very much for highlighting this important potential misunderstanding of the paper (and the title) which we did not foresee. The aspect of the necessity of non-linear reaction terms is indeed fully acknowledged. It is the subject of the companion article providing a step-by-step introduction into the theory of excitable waves stemming from the THMC reaction terms and putting proper reference to the FKPP work and the subsequent work in the Russian literature. This discussion was beyond scope for the first part. The important reference to excitation waves as discussed in the companion paper is now highlighted on line 11, 40 and 353 ff in the revised manuscript.*

*The title was meant to convey the important fact that without cross-diffusion terms no new type of soliton-like (quasi-soliton or cross-diffusion) wave is generated (triggered) given the existence of excitable waves. We appreciate, however, that this point only becomes clear after reading the paper and one could misunderstand the title. We therefore changed the title, added a sentence at the end of the abstract and further explanation to the wave operator discussion on line 353 in*

*the annotated revised manuscript. The wave operator was originally introduced to highlight the role of the non-diagonal terms which were overlooked in our early work on compaction bands.*

*This early work on the standing wave solution of the Korteweg-de Vries (KdV) equation as an explanation for compaction bands in nature considered indeed an extension to McKenzie's theory by using a nonlinear rheology (we obtained analytical solutions with power law exponent 3 reaction term). It resulted in unphysical infinite overpressure spikes on the instabilities. On line 48 following we now highlight this problem of both our early work on the KdV equation and the axiomatic classical localisation theory of plasticity theory by Hill and Rudnicki and Rice. In the new paragraph following line 55  we also discuss the limits of our own earlier work to improve this approach and come back to the important point on line 353.*

*Because of the proposed importance to understand localisation bands in nature our team and our collaborators have since tried to regularise the problem for the purpose of obtaining numerically stable solutions. Generalised equations were analytically not tractable and numerical work was therefore crucial. Several solutions that worked for some cases were found. The intention of Hu et al.(2020) and the present work is to close this ill-posed problem by considering the nonlocal nature of the reaction term.*

*In Hu et al. (2020) we presented pioneering work for HM coupling. The current work is proposing a closure for all considered THMC extensions through generalisation of cross-diffusion terms proposed in Hu et al.(2020). The work also spells out clearly the differences to earlier work in localisation theory based on acceleration waves missing in Hu et al (2020). Finally, this work also attempts to provide the reader a bridge to the rich published literature on quasi-solitons where many more aspects of cross-diffusion waves are discussed in different communities, which should be easily digestible given our introduction. This intention is now highlighted more specifically in various sections of  the revised version to elucidate the important differences, hopefully without distracting from the key ideas presented in this work.*

*This article is indeed all about the non-diagonal terms in the THMC matrix and the consideration of the propagation of soliton-like waves which was not yet clearly addressed before. The interesting aspect is that without off-diagonal terms but with non-linear source terms no cross-diffusion waves are triggered but only solitary waves with Hopf and Turing bifurcations. New work by our team has provided proof of this statement for the  generalised case of poro-mechanics applications. Only by inclusion of cross-diffusion (acknowledging higher order reactive source terms for the solid skeleton and a linear fluid reaction term) excitable waves with quasi-solitonic behaviour are triggered. As our new work is only in draft manuscript form we refer in the revision on line 363  to Tsyganov (2007) who identified the important finding first for the mathematically similar Fitz-Hugh Naguno system.*

*The companion paper is more of a step-by-step introduction as it starts indeed with a summary of the theory of excitable waves and then goes over into a discussion of the influence of the non-diagonal terms in the diffusion matrix.*

Line 42 – the authors may wish to consider a consistent way of referring to the second, companion paper (e.g., Paper 1, and Paper 2)

*Reply: The second paper is now named "companion article" throughout the text.*

Line 79 – omit the second "s" in "systemss"

*Reply: Done*

Line 89 – sentence is confusing ("processes travels"??)

*Reply: it now reads that the information disperses*

Line 95 – omit "s" in details; omit comma after diffusion

*Reply: Done*

Line 109 – omit semi colon and instead use parenthesis to be consistent with earlier terms in the sentence, i.e., "chemical driving forces (chemo-taxis)"

*Reply: Done*

Line 110 – omit "Thermal, Hydrodynamic, Mechanical, and Chemical" as abbreviation has been defined previously

***Reply:*** *Done*

Line 123 – omit comma after "physics"

***Reply:*** *Done*

Line 134 – put references in parentheses.

***Reply:*** *Done*

Line 253 – It is confusing as to what is meant by "former" and "latter" with respect to the previous sentence; please rewrite

***Reply:*** *Done*

Line 276 – the creeping flow assumption ignores the acceleration terms in the Navier-Stokes equation, but there still can be effects of "gravitational acceleration", i.e. creeping flow in a gravity field.

***Reply:*** *Done*

Line 285 – The phrasing "The compatibility condition relating jumps…" is not a complete sentence

***Reply:*** *Done*

Line 293 – I believe you want to make the phrase "It follows from Eq. 7" the start of a new sentence

***Reply:*** *Done*

Line 324 – "tabulated in Table 1. The reaction rates…"?

***Reply:*** *Done*

Line 326 – there is no operator H-hat-sub"THMC" in Table 1.

***Reply:*** *The intention of the introduction of the wave operator, based on the self- diffusivities defined in Table 1, was to build up for the following discussion on the necessity for considering cross-diffusion coefficients from the nonlocal reactive source term. This operator maps the excitation waves into a new space that when including cross-diffusion coefficients generates quasi-solitons. The operator was originally introduced as a mathematically more succinct form of the lengthy step-by-step "convolution wave filter" discussion in the companion article. This information was obviously missing and is now given on line 353ff .*